# Defective NADPH production in mitochondrial disease complex I causes inflammation and cell death

Eduardo Balsa[1,2,4], Elizabeth A. Perry[1,2], Christopher F. Bennett[1,2], Mark Jedrychowski[2], Steven P. Gygi[2], John G. Doench [3] & Pere Puigserver[1,2,3 ✉]

Electron transport chain (ETC) defects occurring from mitochondrial disease mutations compromise ATP synthesis and render cells vulnerable to nutrient and oxidative stress conditions. This bioenergetic failure is thought to underlie pathologies associated with mitochondrial diseases. However, the precise metabolic processes resulting from a defective mitochondrial ETC that compromise cell viability under stress conditions are not entirely understood. We design a whole genome gain-of-function CRISPR activation screen using human mitochondrial disease complex I (CI) mutant cells to identify genes whose increased function rescue glucose restriction-induced cell death. The top hit of the screen is the cytosolic Malic Enzyme (ME1), that is sufficient to enable survival and proliferation of CI mutant cells under nutrient stress conditions. Unexpectedly, this metabolic rescue is independent of increased ATP synthesis through glycolysis or oxidative phosphorylation, but dependent on ME1-produced NADPH and glutathione (GSH). Survival upon nutrient stress or pentose phosphate pathway (PPP) inhibition depends on compensatory NADPH production through the mitochondrial one-carbon metabolism that is severely compromised in CI mutant cells. Importantly, this defective CI-dependent decrease in mitochondrial NADPH production pathway or genetic ablation of SHMT2 causes strong increases in inflammatory cytokine signatures associated with redox dependent induction of ASK1 and activation of stress kinases p38 and JNK. These studies find that a major defect of CI deficiencies is decreased mitochondrial one-carbon NADPH production that is associated with increased inflammation and cell death.

[1] Department of Cancer Biology, Dana-Farber Cancer Institute, Boston, MA, USA. [2] Department of Cell Biology, Harvard Medical School, Boston, MA, USA. [3] Broad Institute of MIT and Harvard, Cambridge, MA, USA. [4] Present address: Department of Biochemistry, Universidad Autónoma de Madrid (UAM) and Instituto de Investigaciones Biomédicas Alberto Sols (CSIC-UAM), Madrid, Spain. ✉email: pere_puigserver@dfci.harvard.edu

Mitochondrial oxidative phosphorylation (OXPHOS) synthesizes ATP to maintain cellular bioenergetics, particularly under restricted glycolysis or increased energetic demand[1,2]. Electron transport chain (ETC) defects occurring from mitochondrial disease mutations compromise cell viability causing diverse pathologies like neurodegeneration, myopathies, optic atrophy, or deafness[3–6]. However, the precise metabolic processes altered by a defective mitochondrial ETC that drive these fatal disorders are not entirely understood. In addition to ATP synthesis, mitochondria provide a wide range of critical metabolic-dependent functions that include cellular redox balance, calcium homeostasis, inflammatory signals, and apoptosis[7]. Furthermore, complex I (CI)-driven reoxidation of NADH to NAD$^+$ is a critical ETC function that will unbalance metabolic processes through alterations in mitochondrial NAD$^+$/NADH ratios. Interestingly, interventions aimed to correct these ratios such as boosting NAD$^+$ levels have proven to be beneficial in diseases associated with mitochondrial dysfunction[8,9]. In this redox balance, the mitochondrial matrix harbors the one-carbon metabolism pathway that, similar to the cytosolic, consumes serine and tetrahydrofolate through a series of redox reactions and produces formate, which upon export into the cytosol feeds into the pathway for purine and methyl group biosynthesis[10]. Mitochondrial one-carbon metabolism produces cellular NADPH which is coupled to the ALDH1L2 enzymatic activity[11]. Interestingly, increased levels of mitochondrial one-carbon metabolism enzymes have been found in different models of disrupted ETC whose expression is regulated by the activating transcription factor 4 (ATF4)[12,13].

The design of cell-based assays to screen for genes that rescue detrimental phenotypes associated with ETC inhibition has been particularly challenging. This difficulty stems from the highly glycolytic metabolic signature that most of the cell lines exhibit in culture conditions. As a consequence, defects in OXPHOS are well tolerated and masked by glycolytic ATP production[2]. To overcome this difficulty, we have recently designed cell-based screening platforms culturing OXPHOS-deficient cells in galactose media[14]. Upon these nutrient stress conditions, cells are forced to fully rely on mitochondrial oxidative metabolism for survival[15]. In this study, we employ a genome-wide clustered regularly interspaced short palindromic repeat (CRISPR)/Cas9 activator library to identify genes that could prevent cell death of CI mutant cells cultured in galactose. One of the top positive hits is the cytosolic malic enzyme 1 (ME1) that enables survival and proliferation of CI-deficient cells under nutrient stress conditions. We find that ME1-mediated survival is not dependent on increasing ATP levels but rather on restoring defective NADPH production. Wild-type (WT), but not CI mutant, cells increase mitochondrial one-carbon metabolism in galactose conditions to compensate for the decreased pentose phosphate pathway (PPP)-dependent NADPH production. As a consequence, CI mutant cells exhibit decreased levels of NADPH and glutathione (GSH), with a concomitant increase in oxidative stress that leads to inflammation and cell death. These results show an unanticipated metabolic mechanism whereby ETC function is linked to NADPH homeostasis and protects against oxidative stress and inflammation, particularly under nutrient stress conditions.

## Results

### Gain-of-function CRISPR/Cas9 screen identifies ME1.
Mitochondrial mutations in genes encoding for OXPHOS components reduce cellular fitness, including survival, which is exacerbated under stress conditions[16,17]. However, the metabolic processes that directly cause cellular damage and pathologies in human mitochondrial disease mutations are not entirely understood. To gain new insights into this metabolic problem, we designed a gain-of-function whole-genome CRISPR-based positive screen to identify genes whose increased expression protected human mitochondrial disease CI mutant ND1 (3796A>G) cybrid cells from galactose-induced cell death. We employed a validated pooled lentiviral single guide RNA (sgRNA) library targeting the promoter of 18,675 genes in the human genome[18]. Each gene activated by this library is targeted by 3–6 guides (sgRNAs) that are split among two half-libraries. ND1 mutant cells, stably expressing a nuclease-deactivated Cas9 (dCas9) fused to a transcriptional activator domain (VP64), were infected with lentiviral libraries and selected for 1 week. Cells were then challenged for two rounds of galactose before surviving cells were collected for sequencing (Fig. 1a). For each gene, we calculated log2 fold change values in the abundance of the sgRNAs targeting the gene. As expected, most genes, as well as the control sgRNAs, scored negative with just a few genes scoring double positive in both library replicate sets (Fig. 1b). The inability of OXPHOS-deficient cells to survive under galactose has been attributed to the slow enzymatic conversion of galactose into glucose and limiting glycolytic-dependent ATP production[19]. The fact that galactose-1-phosphate uridyl transferase was one of the top hits indicates that this enzymatic reaction is a limiting step process and further validated the whole-genome screen platform (Fig. 1b). Interestingly, BCL-2 family members were among the strongest hits, revealing that apoptosis is the cell death mechanism under these nutrient stress conditions. ME1, an enzyme that catalyzes the reversible oxidative decarboxylation of malate to pyruvate, was the best scoring gene when ranked by significance (Fig. 1c). Further validation using two independent guides confirmed that ND1 cells were able to survive and proliferate in galactose conditions after ME1 induction (Fig. 1d, e). ME1-induced resistance occurred in the absence of pyruvate suggesting that the enzymatic reaction favors the conversion of malate into pyruvate with concomitant production of NADPH (Fig. 1f, g). A similar phenotype was observed in CI-deficient cells with a pathological mutation in ND6 (14459G>A) (Supplementary Fig. 1a–c) and in multiple cell lines when CI was inhibited pharmacologically (Supplementary Fig. 1d). Conversely, ME1 ablation sensitized WT cybrid cells to galactose-induced cell death (Supplementary Fig. 1e). Of note, ME1 rescued CI-deficient cells without increasing mitochondrial respiration or ETC components (Supplementary Fig. 1f–h).

### ME1 favors reductive carboxylation of glutamine.
Next, we investigated how ME1 rewired substrate utilization. When glucose is limiting, glutamine becomes the primary substrate to support the mitochondrial tricarboxylic acid (TCA) cycle, and increased glutamine utilization is a metabolic hallmark of cells with ETC dysfunction[20,21]. Malate, the substrate of the ME1, can be generated by glutamine through the oxidative pathway or reductive carboxylation of glutamine-derived α-ketoglutarate (α-KG) (Fig. 2a). To determine how ME1 controls glutamine utilization, sgNeg and sgME1 ND1 mutant cells were incubated for 3 h in galactose media supplemented with $^{13}$C-labeled ([U-$^{13}$C$_5$]) glutamine. Nearly 78% of the glutamine-derived malate was already labeled after 3 h (Fig. 2b). ME1 overexpression increased malate formation from glutamine-reductive metabolism (M + 3) by 17% while decreasing malate M + 4 and overall oxidation of glutamine by 19% (Fig. 2c–e). Increasing supplementation of malate, however, did not result in cell survival rescue suggesting that protein levels or activity of the enzyme rather than substrate availability underlie these beneficial effects (Fig. 2f). These results suggest that increased ME1 expression in glucose-restricted CI mutant cells promoted glutamine flux through the mitochondrial/cytoplasmic reductive pathway.

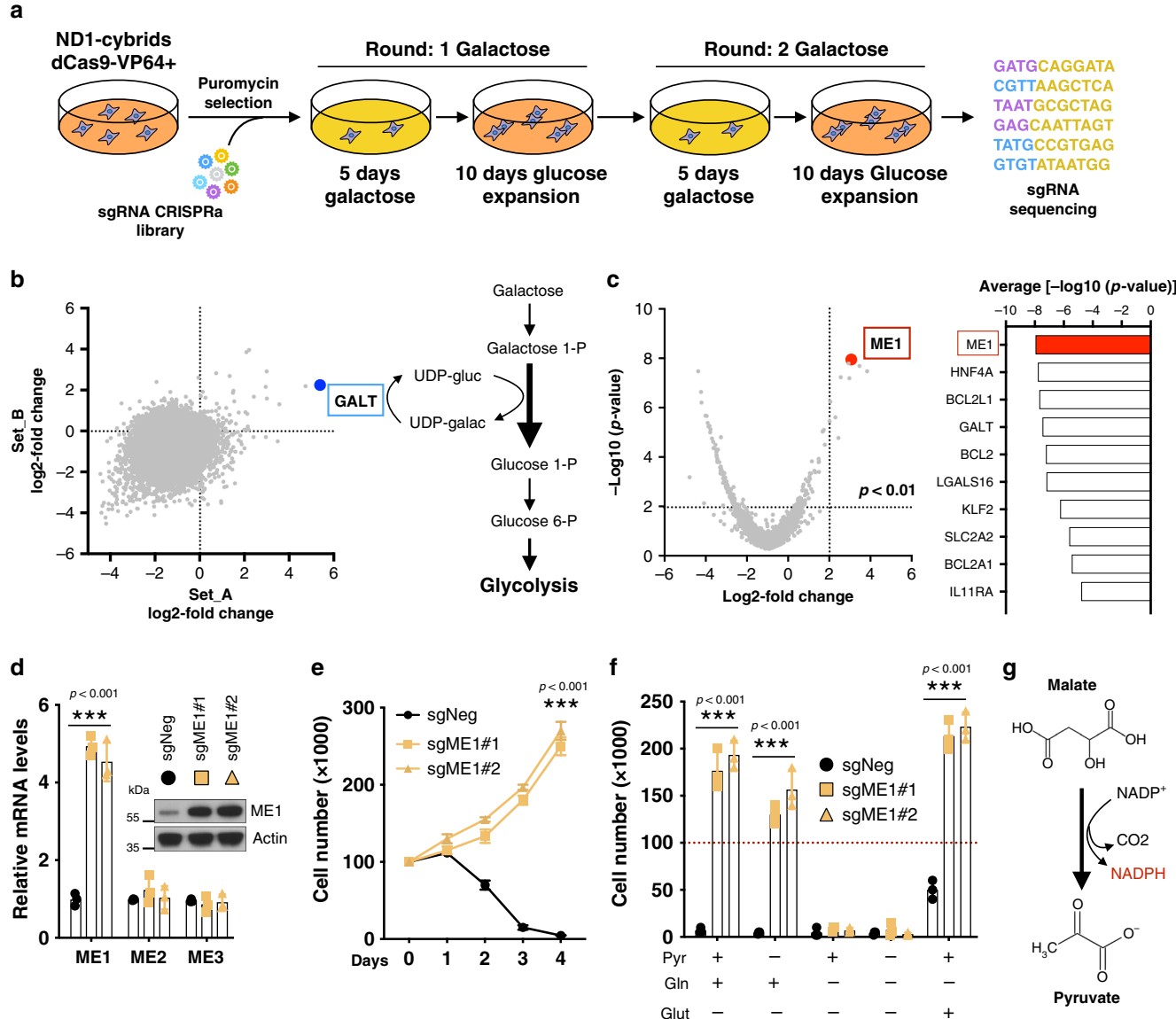

**Fig. 1 Identification of ME1 through a gain-of-function CRISPR/Cas9 screen. a** Schematic overview of the genome-wide CRISPR activator screen. **b** Scatterplot showing gene enrichment after galactose challenge using two independent library replicate sets . Highlighted in blue is GALT gene, which represents the rate-limiting enzyme in galactose-to-glucose conversion. **c** Volcano plot highlighting ME1 (red dot) as the best scoring gene (left) and the top 10 scoring genes ranked by significance. **d** Specific mRNA and protein induction of ME1 using two different guides in ND1 mutant cells ($n = 3$). **e** Cell survival and proliferation curves in ME1-overexpressing ND1 mutant cells under galactose ($n = 3$). **f** Cell survival in ME1-overexpressing ND1 mutant cells cultured under galactose media for 96 h in the presence or absence of glutamine (4 mM), glutamate (4 mM), or pyruvate (1 mM) ($n = 3$). **g** ME1 catalyzes the decarboxylation of malate to pyruvate generating NADPH. Immunoblots shown are representative of >3 independent experiments, and all other experiments are represented as means ± SEM., $n > 3$ biological replicates. Asterisks denote *$p < 0.05$, **$p < 0.01$, or ***$p < 0.001$. Paired two-tailed Student's $t$ test in **d**, **e** and two-way ANOVA in **f**. Pyr pyruvate, Gln glutamine, Glut glutamate. Red dashed lines indicate initial seeding density.

**Impaired NADPH and GSH levels in mitochondrial mutant cells lead to oxidative stress**. Since ME1 is a NADPH-generating enzyme[22], we sought to determine whether NADPH levels were linked to survival in ND1 cells cultured in glucose-restricted conditions. NADPH levels as well as NADPH/NADP$^+$ ratios were markedly reduced in ND1 mutant cells and were restored by ME1 overexpression (Fig. 3a, b). Reduced NADPH translated into lower GSH levels and significant increases in oxidative stress that was ameliorated by ME1 overexpression (Fig. 3c, d). To assess whether antioxidants promoted cell survival, ND1 mutant cells were supplemented with GSH, N-acetyl cysteine (NAC), or MitoQ. Both GSH and its precursor NAC were able to rescue cell

death although only GSH displayed a robust and long-lasting effect (Fig. 3e, f). Surprisingly, MitoQ, which specifically buffers mitochondrial-generated reactive oxygen species (ROS)[23], did not rescue cell survival (Fig. 3e), suggesting that cytosolic oxidative stress rather than mitochondrial-produced ROS might initiate cell death. The PPP represents an important source of cytosolic NADPH[24]. We then tested whether pharmacological PPP inhibition might recapitulate the phenotypes observed in galactose-grown cells that exhibit reduced PPP activity. Notably, ND1 mutant, but not WT cells, had increased oxidative stress and cell death after pharmacological PPP inhibition that is ameliorated when ME1 is expressed (Fig. 3g–i). Consistent with ME1-

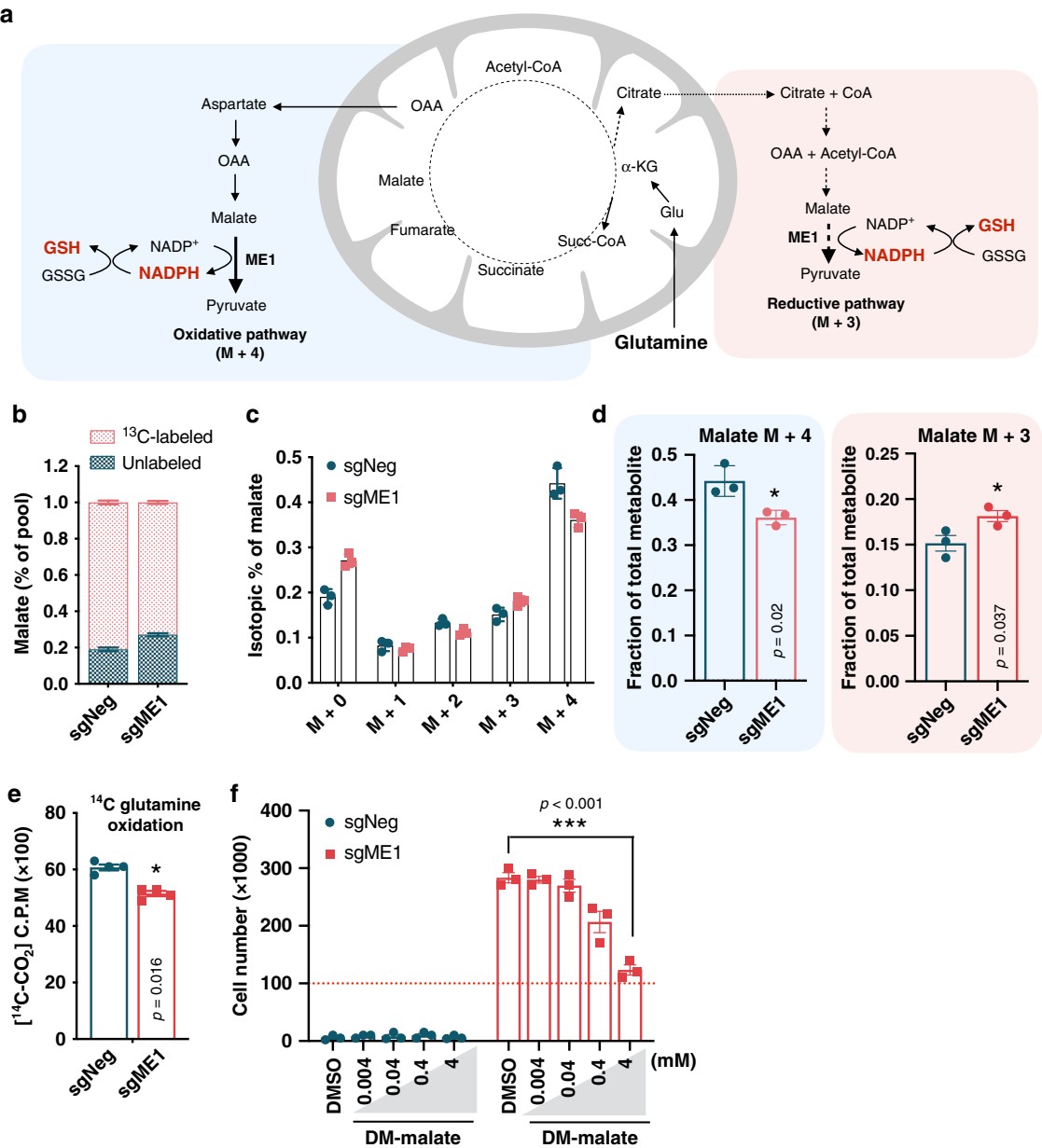

**Fig. 2 ME1 induction promotes reductive carboxylation of glutamine. a** Model illustrating the fate of fully labeled $^{13}C$ glutamine after entering the TCA cycle. Glutamine oxidation generates $M + 4$ labeled substrates while its reductive carboxylation generates $M + 3$ labeled substrates. Note that ME1 activity is coupled to NADPH production and reduction of oxidized glutathione. **b** Percentage of labeled and unlabeled malate in ND1 mutant cells after 3 h incubation with $^{13}C$-labeled ([U-$^{13}C_5$]) glutamine ($n = 3$). **c** Isotopomer distribution of malate in sgNeg and sgME1 ND1 mutant cells cultured in the presence of $^{13}C$ glutamine for 3 h ($n = 3$). **d** ME1 overexpression decreases malate $M + 4$ originated by oxidation of glutamine in the TCA cycle and increases malate $M + 3$ coming from reductive carboxylation of glutamine ($n = 3$). **e** $^{14}C$ glutamine oxidation is reduced in ND1 mutant cells overexpressing ME1 ($n = 3$). **f** Supplementation of cell permeable dimethyl-malate (DM-malate), at the indicated doses, did not increase survival of ND1 under galactose conditions ($n = 3$). Experiments are represented as means ± SEM., $n > 3$ biological replicates. Asterisks denote *$p < 0.05$, **$p < 0.01$, or ***$p < 0.001$. Paired two-tailed Student's $t$ test in **d**, **e** and one-way ANOVA in **f**. Red dashed lines indicate initial seeding density.

dependent cell survival, GSH treatment protected against PPP inhibition-mediated cell death in different cell lines with severe ETC defects (Supplementary Fig. 2a). These results indicate that CI inhibition under nutrient stress, similar to PPP reduction, decreases NADPH levels and causes oxidative stress-dependent cell death that can be rescued by cytosolic ME1 or exogenous supplementation of GSH.

**OXPHOS dysfunction impairs one-carbon metabolism and sensitizes CI mutant cells to oxidative stress.** To address the

cause of the different sensitivities to nutrient stress-induced cell death between WT and ND1 mutant cells, we performed metabolomic analysis. Whereas both cell types exhibited similar decreases in glycolytic and PPP intermediates in galactose conditions (Supplementary Fig. 2b), WT cells were protected from NADPH and GSH depletion at subsequent cell death. To identify the metabolic pathways that maintain NADPH and GSH levels during these nutrient stress conditions in WT cells, we surveyed for metabolites that were differentially regulated between WT versus ND1 mutant cells in galactose conditions. We identified

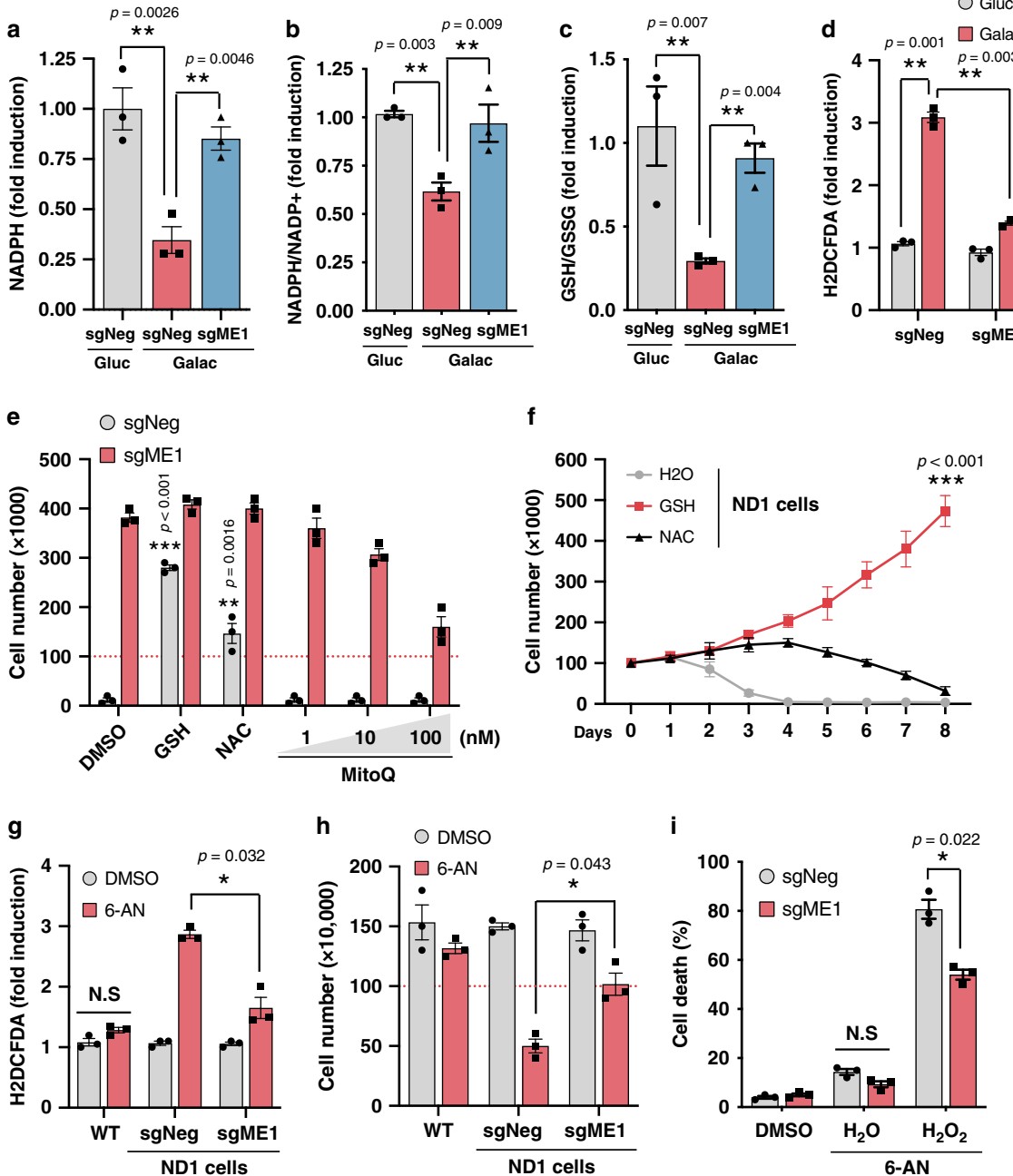

**Fig. 3 Complex I disease mutations reduce levels of NADPH and GSH and cause oxidative stress. a** Relative NADPH levels analyzed by LC-MS, **b** NADPH/NADP$^+$ ratio, **c** GSH/GSSG ratio, and **d** relative reactive oxygen species (ROS) levels measured using dichlorodihydrofluorescein diacetate (H2DCFDA) in sgNeg and sgME1 ND1 mutant cells cultured under glucose or galactose conditions for 48 h ($n = 3$). **e** Cell number of galactose-grown ND1 cells (96 h) supplemented with GSH (2 mM), NAC (4 mM), or the indicated doses of MitoQ ($n = 3$). **f–i** Growth curves in galactose-grown ND1 cells treated with GSH or NAC. 6-Aminonicotinamide (6-AN) (100 μM) was used to inhibit PPP ($n = 3$), and **g** ROS levels, **h** cell number, and **i** H$_2$O$_2$-induced cell death was assessed after 48 h in WT cells ($n = 3$). Experiments are represented as means ± SEM., $n > 3$ biological replicates. Asterisks denote *$p <$ 0.05, **$p < 0.01$, or ***$p < 0.001$. Two-way ANOVA in **a–d**, **g–i** and paired two-tailed Student's $t$ test in **e**, **f**. Gluc glucose, Galac galactose. Red dashed lines indicate initial seeding density.

decreases in serine and 5,10-methylenetetrahydrofolate in WT cells, whereas these metabolites accumulated in ND1 mutants (Fig. 4a). This suggested that one-carbon metabolism is compromised in cells with impaired ETC function. In-depth proteomic analysis of WT cells under galactose conditions revealed significant increases in protein components of mitochondrial one-carbon metabolism and serine biosynthesis (Fig. 4b). To directly assess the activity of this pathway, we measured formate production in isolated mitochondria that were

incubated with serine and ADP + Pi to stimulate respiration. Notably, pharmacological inhibition of CI in WT or ND1 mutant cells showed a strong reduction in serine-derived formate levels, comparable to SHMT2 CRISPR-depleted cells (Fig. 4c) revealing that mitochondrial one-carbon activity is linked to ETC function. One of the potential mechanisms whereby mitochondrial one-carbon metabolism might be deficient is the fact that MTHFD2/L uses NAD$^+$ and therefore could be sensitive to a reduced NAD$^+$/NADH ratio from CI inhibition. To test this, we took advantage

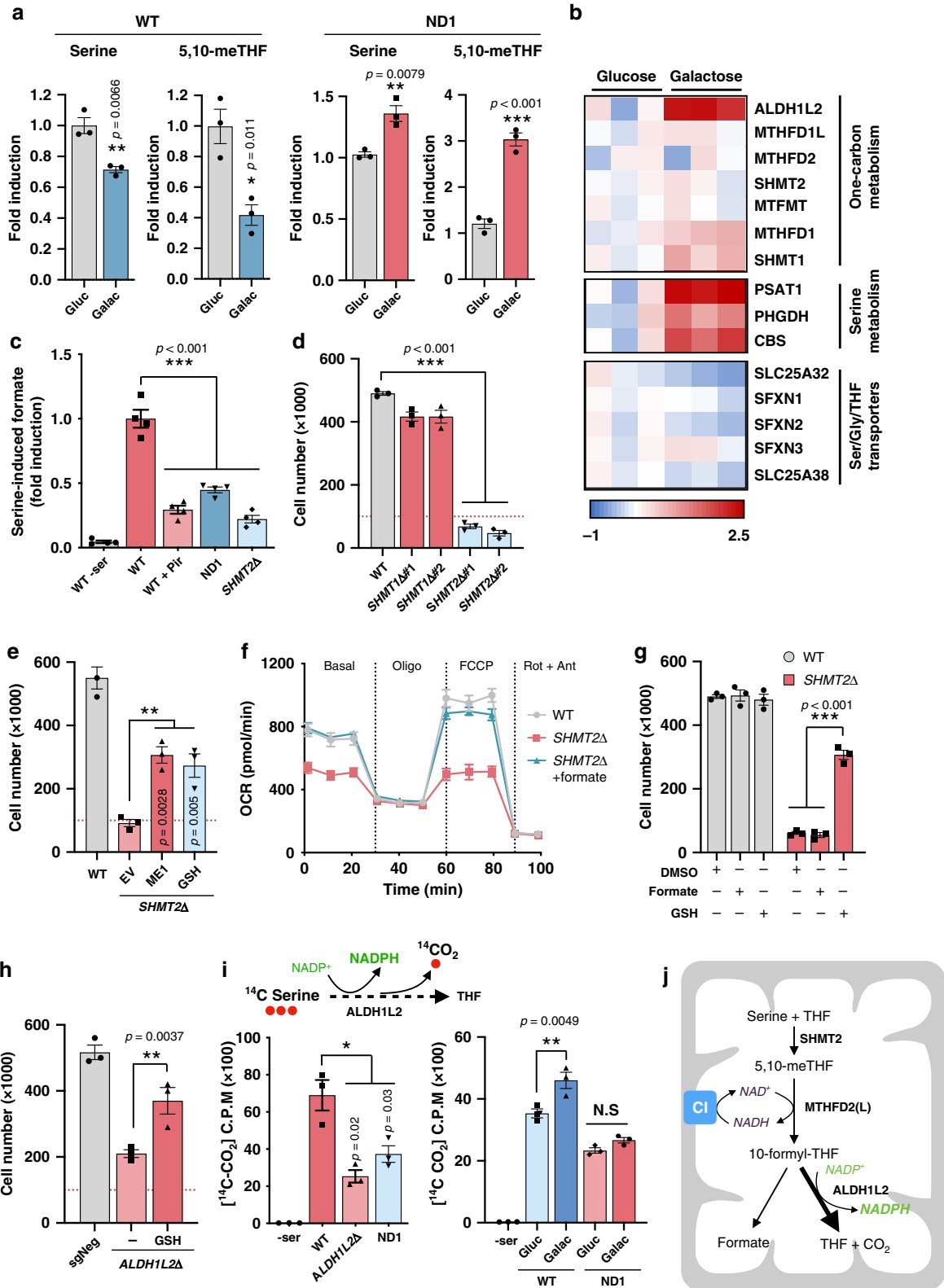

of the NADH oxidase from *Lactobacillus brevis* (LbNOX), which has been shown to regenerate NAD+ in cells with disrupted ETC by selectively consuming NADH[8]. Consistent with an imbalanced NAD+/NADH ratio constraining the flux of mitochondrial one-carbon metabolism, mitochondrial-targeted LbNOX was able to restore serine-derived formate in ND1 mutant cells (Supplementary Fig. 2c). One-carbon metabolism occurs both in the

cytoplasm and the mitochondria, with parallel reactions taking place in a highly compartmentalized manner[10,25]. We observed that suppression of mitochondrial (SHMT2 CRISPR cells), but not cytosolic (SHMT1 CRISPR cells) one-carbon metabolism, sensitized WT cells, similar to CI-defective cells, to galactose or PPP inhibition-mediated cell death (Fig. 4d and Supplementary Fig. 2d–f). Increased ME1 expression or GSH supplementation

**Fig. 4 Mitochondrial one-carbon metabolism is linked to functional ETC activity. a** Relative levels of intracellular serine and 5,10-methylenetetrahydrofolate (5,10-meTHF) in WT and ND1 mutant cells cultured either in glucose or galactose for 24 h and analyzed by LC-MS ($n = 3$). **b** Proteomics heatmap in WT cells exhibiting relative expression (log2 fold change) of proteins differentially regulated under 48 h galactose. **c** Measurement of formate production from serine using isolated mitochondria from WT, ND1, or SHMT2Δ cells ($n = 4$). **d** Cell number of WT, SHMT1Δ, and SHMT2Δ cell culture in galactose for 96 h ($n = 3$). **e** ME1 overexpression and 2 mM GSH supplementation rescues cell survival in galactose-grown SHMT2Δ cells ($n = 3$). **f** Seahorse analysis in WT sgNeg and SHMT2Δ in the absence or presence of 1 mM formate ($n = 5$). **g** GSH but not formate rescued cell survival in SHMT2Δ cells ($n = 3$). **h** GSH rescued cell number in ALDH1L2Δ cells cultured in galactose ($n = 3$). **i** ALDH1L2 converts 10-formyltetrahydrofolate to tetrahydrofolate and carbon dioxide in an $NADP^+$-dependent reaction. ND1 cells display decreased serine-derived $CO_2$ release compared to WT cells. ALDH1L2Δ cells were used as positive control (left panel). Galactose stimulated serine-derived $CO_2$ release in WT cells but not in CI-deficient ND1 mutant cells (right panel) ($n = 3$). **j** Model illustrating the dependency of mitochondrial one-carbon metabolism on ETC function for NADPH production and how upregulation of ALDH1L2 stimulates NAPDH production in glucose-free conditions. Immunoblots shown are representative of >3 independent experiments, and all other experiments are represented as means ± SEM, $n > 3$ biological replicates. Asterisks denote *$p < 0.05$, **$p < 0.01$, or ***$p < 0.001$. Paired two-tailed Student's $t$ test in **a** and two-way ANOVA in **c–j**. Gluc glucose, Galac galactose, ser serine, Gly glycine, THF tetrahydrofolate, Pir Piericidin, Oligo oligomycin, Rot rotenone, Ant antimycin A. EV denotes empty vector. Red dashed lines indicate initial seeding density.

rescued the cell death phenotype in SHMT2 CRISPR cells (Fig. 4e and Supplementary Fig. 2g). Loss of SHMT2 has been shown to reduce mitochondrial protein translation due to depletion of formylmethionyl-tRNAs[26]. Indeed, SHMT2 knockout (KO) cells displayed reduced respiration and decreased levels of MT-COI, a mitochondrial encoded protein. Addition of exogenous formate to cell culture media restored defects in mitochondrial respiration and mitochondrially encoded protein MT-COI (Fig. 4f and Supplementary Fig. 2h). However, formate was not sufficient to prevent cell death in SHMT2-deficient cells (Fig. 4g) suggesting that increased energetics is insufficient to rescue cell death under these conditions. The one-carbon metabolism enzyme ALDH1L2 is one of the main contributors to mitochondrial NADPH production[27]. Notably, this enzyme was highly upregulated in galactose conditions when analyzed by mass spectrometry proteomics and western blot analysis (Fig. 4b and Supplementary Fig. 3a). In fact, ALDH1L2 depletion rendered WT cells sensitive to cell death after glucose removal, which was rescued by GSH supplementation (Fig. 4h and Supplementary Fig. 3b). Because ALDH1L2 enzymatic reaction produces equimolar levels of $CO_2$ and NADPH, we measured serine-derived $CO_2$ as a reliable indicator of mitochondrial NADPH production. In agreement with formate assays, isolated mitochondria from ND1 cells incubated with [3-$^{14}$C] serine displayed reduced serine C3-derived $CO_2$ production, similar to ALDH1L2 KO cells. Moreover, ND1 mitochondria, in contrast to WT, failed to increase $CO_2$ production under galactose conditions, indicating that ETC failures impair one-carbon oxidation and NADPH production (Fig. 4i). Taken together, these results suggest that, under nutrient conditions that limit glucose availability, increased serine-driven mitochondrial one-carbon metabolism flux is preferentially channeled toward NADPH production for reductive/oxidative balance rather than preserving carbon blocks for nucleotide synthesis and proliferation (Fig. 4j). Thus, in cells with impaired mitochondrial one-carbon metabolism, similar to CI mutant cells, increased cytosolic oxidative stress causes cell death.

NADPH is a compartmentalized molecule unable to freely diffuse through lipid membranes. Therefore, one of the questions is how WT cells under glucose-restricted conditions are able to transfer mitochondrial-produced NADPH to the cytosol, a process that is impaired in CI mutant cells. Within the TCA cycle, NADPH can be used as a cofactor for the mitochondrial $NADP^+$-dependent isocitrate dehydrogenase 2 (IDH2) to reductively carboxylate α-KG to isocitrate, which in turn can be converted to citrate and be exported to cytosol. Cytosolic isocitrate can then be oxidatively decarboxylated by cytosolic $NADP^+$-dependent IDH1, producing cytosolic NADPH to fuel reduction of glutathione disulfide. Interestingly, we observed that IDH1, and to a lower extent IDH2, were markedly upregulated

upon galactose treatment in our proteomic analysis and verified by western blot (Supplementary Fig. 3c). Genetic ablation of IDH1 sensitized WT cells to galactose media and was rescued by GSH supplementation (Supplementary Fig. 3d). Furthermore, we observed similar galactose cell-death sensitivity when IDH1-R132H and IDH2-R172K mutant forms were overexpressed in WT cells. These mutations change the function of the enzymes, causing them to produce 2-hydroxyglutarate at the expense of consuming NADPH during the process, thus creating a futile cycle where WT-IDH1/2-produced NADPH is downstream exhausted by mutant IDH1/2 (Supplementary Fig. 3e). This indicates that, in WT cells, but not in CI mutant cells, the mitochondrial NADPH shuttle system involving IDHs IDH2 and IDH1 could be responsible for transferring reducing equivalents from the mitochondria to the cytosol (Supplementary Fig. 3f), thus providing additional reducing power and alleviating cytosolic oxidative stress in conditions of diminished PPP activity.

**Oxidative stress in mitochondrial CI and one-carbon metabolism-deficient cells underlies inflammation in vitro and in vivo.** To gain more mechanistic insights into the unbalanced reductive/oxidation processes that cause cell death, we further analyzed the positive hits from the CRISPR screen in CI mutant cells (Fig. 1c) to identify genes involved in oxidative stress. Among these hits, peroxiredoxin 1 (PRDX1), a member of the peroxiredoxin family of antioxidant enzymes, was found as a positive hit and its overexpression mildly increased survival in ND1 mutant cells (Supplementary Fig. 4a). PRDX1 has been found to interact and participate in the activation of the apoptosis signaling kinase 1 (ASK1), a redox-sensitive kinase that mediates cell stress signaling through downstream phosphorylation of mitogen-activated protein kinases (MAPKs) p38 and c-Jun N-terminal kinase (JNK)[28]. Consistent with this finding, galactose treatment or PPP inhibition activated this signaling cascade in ND1 mutant cells but not in WT counterparts (Fig. 5a, b). In addition, activation of this kinase stress signaling module in ND1 mutant cells, but not in WT, strongly induced a pro-inflammatory gene expression signature including cytokines (Fig. 5c, d). Remarkably, ameliorating oxidative stress either by increased NADPH production through ME1 overexpression (Fig. 5c, d) or GSH treatment (Supplementary Fig. 4b, c) strongly suppressed this inflammatory response. Moreover, and in alignment with the cell death results (Fig. 4d), SHMT2 CRISPR cells, defective in mitochondrial one-carbon metabolism, also showed significant activation of the same inflammatory markers (Supplementary Fig. 4d). These results indicate that NADPH deficiency triggered by CI-dependent inhibition of mitochondrial one-carbon metabolism causes inflammatory gene signature.

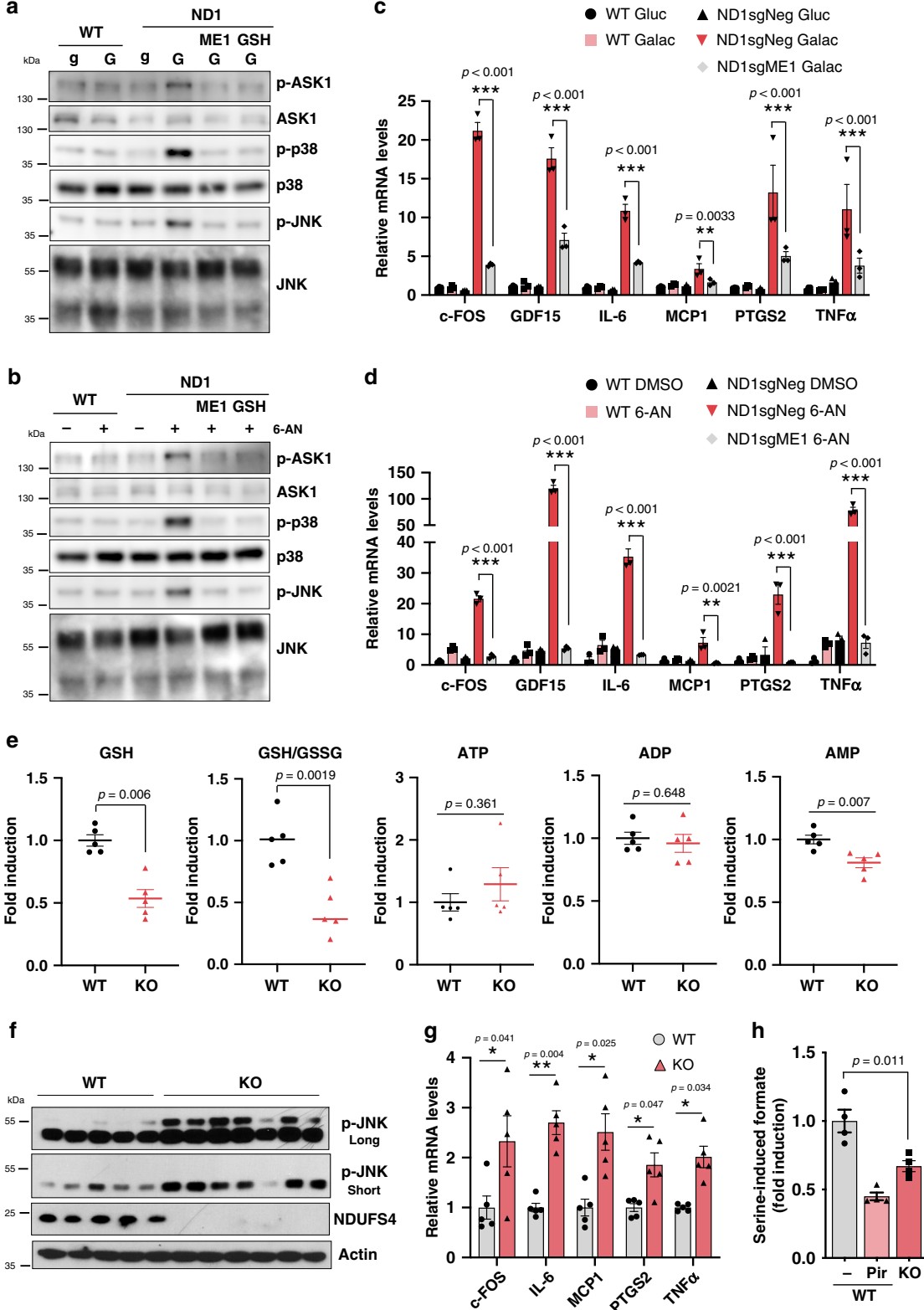

Next, we determined whether the unbalanced reductive/oxidative metabolism and inflammatory response observed in mitochondrial mutant cells also occurred in vivo. We conducted metabolomic analysis using the brains of *Ndufs4* KO mice, a model of CI deficiency that develops a fatal neuropathy and recapitulates human Leigh syndrome[3,29]. Consistent with our results in vitro, brain metabolomic analysis of *Ndufs4* KO mice exhibited significant decrease in NADPH and GSH with apparent

normal levels of ATP (Fig. 5e and Supplementary Fig. 4e). This metabolic signature correlated with activation of JNK pathway and increased expression of inflammatory markers (Fig. 5f, g). In addition, isolated brain mitochondria from *Ndufs4* KO mice also exhibited a significant reduction in serine-driven mitochondrial one-carbon metabolism that is in alignment with decreased NADPH levels (Fig. 5h). These results indicate that a major metabolic defect in human CI disease mutations is an impairment

**Fig. 5 Complex I inhibition, in vitro and in vivo, is associated with an inflammatory gene response caused by increased oxidative stress.** Immunoblots showing oxidative stress-mediated activation of ASK1/P38/JNK axis specifically in ND1 mutant cells **a** cultured in galactose or **b** after PPP inhibition using 6-AN at 100 μM for 48 h. Pro-inflammatory gene expression signature is induced in **c** 48 h galactose-grown or **d** PPP-inhibited ND1 cells and rescue by ME1 overexpression (n = 3). **e** Metabolomic analysis in brain samples of WT and *Ndufs4* KO mice. Note that levels of GSH are reduced, while no changes are observed in ATP. GSH/GSSG ratio for WT is 4.3920 +/− 0.9039 (Average +/− Standard) (n = 5). **f** Increased phosphorylation of JNK (including long and short exposures) and **g** induced gene expression of inflammatory markers in the brain of *Ndufs4* KO mice. **h** Reduced formate production using brain-isolated mitochondria from WT or *Ndufs4* KO mice (n = 4). Immunoblots shown are representative of >3 independent experiments, and all other experiments are represented as means ± SEM., n > 3 biological replicates. Asterisks denote *$p < 0.05$, **$p < 0.01$, or ***$p < 0.001$. Two-way ANOVA in **c**, **d**, **h**. Paired two-tailed Student's *t* test in **e**, **g** and one-way ANOVA in **h**. gluc/g glucose, Galac/G galactose, Pir Piericidin.

in mitochondrial-dependent NADPH production that causes inflammation and cell death, without declines in energetics linked to ATP.

Together, these studies show that unbalanced cytosolic reduction/oxidation caused by NADPH deficiency, but not by mitochondrial ROS, underlies the inflammatory and cell death phenotypes caused by nutrient or oxidative stress in mitochondrial disease CI mutations.

## Discussion

Recent studies have failed to identify cytosolic NADPH producers beyond PPP, ME1, and IDH1[24], concluding that these metabolic pathways constitute the three pillars that sustain NADPH homeostasis. Here we report that limited glucose availability suppresses PPP activity, which render cells completely dependent on ME1 and IDH1 for NADPH production. Under these stress conditions, cells adapt by increasing serine catabolism through the mitochondrial one-carbon metabolism to generate NADPH. Our results suggest that mitochondrial-produced NADPH can be transferred to the cytosol via IDH2/IDH1 shuttle, thus establishing a functional link between one-carbon metabolism and the IDH isoforms (also upregulated in glucose-free conditions). CI dysfunction, by compromising one-carbon metabolism, virtually impairs IDH1's ability to extract reducing power from the mitochondria, under nutrient stress conditions, which ultimately leads to exacerbated oxidative stress and cell death. Under these conditions, we found that augmenting ME1 activity was able to compensate for this detrimental loss in NADPH and restore redox homeostasis, promoting cell survival and alleviating an inflammatory phenotype (Fig. 6). We predict that, in the background of mitochondrial dysfunction, cells and tissues with less capacity to compensate via upregulation of PPP or ME1 would be more susceptible to oxidative stress and cellular deterioration.

Interestingly, cell proliferation upon ETC inhibition has been linked to GOT1/2-dependent aspartate activity that increases nucleotide synthesis[30,31], highlighting the importance of functional ETC in cellular processes beyond ATP production to maintain cell fitness and growth. ETC inhibition impairs NADPH production from mitochondrial one-carbon metabolism, which relies on functional CI activity. This pathway is essential for cell viability in conditions of nutrient stress such as glucose deprivation or decreased PPP activity that lower cellular NADPH. Notably, we show that cells harboring mitochondrial CI disease mutations, similar to deficiencies in mitochondrial one-carbon metabolism, are vulnerable to PPP inhibition. Interestingly, mutations that produce glucose-6-phosphate dehydrogenase (G6PD) deficiency in humans result in hemolytic anemia[32]. Even though this enzyme is active in virtually all types of cells, its deficiency specifically damages erythrocytes, literally the only cell in the body that does not contain mitochondria[33]. Based on our studies, it is conceivable that G6PD deficiency will sensitize cells with mitochondrial disease mutations or to insults that impair ETC activity.

Serine synthesis and mitochondrial one-carbon metabolism genes are ATF4 transcription factor targets that are increased in mitochondrial dysfunction mouse models, suggesting an attempt to normalize this inhibited pathway that maintains NADPH balance[12,34]. Our results suggest that functional restoration of mitochondrial one-carbon metabolism in diseases associated with ETC dysfunction might be a potential therapeutic target maintaining NADPH levels, especially under oxidative stress conditions. For example, serine fluxes through mitochondrial one-carbon metabolism are controlled through the $NAD^+/NADH$ ratio that depends on CI activity[35]. Consistent with this, oral administration of nicotinamide riboside, a vitamin B3 and $NAD^+$ precursor, prevents development and progression of mitochondrial myopathy in mice[9]. Along these lines, overexpressing one-carbon metabolism genes in a Drosophila model of Parkinson's disease with mitochondrial defects but intact ETC rescued the loss of dopaminergic neurons[36]. Alternatively, approaches that increase NADPH levels and suppress oxidative stress, such as ME1 expression, could be beneficial and a potential treatment strategy for diseases associated with mitochondrial CI deficiency.

One-carbon metabolism has been extensively studied in the context of cancer since rapidly proliferating cells exploit this pathway for nucleotide biosynthesis to support cell growth[37]. However, its impact on NADPH generation and protection against oxidative stress has been less explored. Here we uncovered a novel link between mitochondrial one-carbon metabolism and inflammation that could have important implications for human diseases. Even though loss of SHMT2 leads to defective OXPHOS due to impaired mitochondrial translation, we show that the cell death and inflammatory phenotype is driven by decreased NADPH production and not mitochondrial translation inhibition.

Previous studies have linked mitochondrial failures and inflammatory pathways associated with mitochondrial DNA leaking to the cytoplasm and activation of the cGAS/STING signaling[38,39]. It would be of interest to determine in what conditions, or specific mitochondrial defects, the different inflammatory cascades, including NADPH deficiency, might contribute to different pathologies. Recent studies have showed that hypoxia or rapamycin treatment can improve the lifespan of Ndufs4 KO mice[40,41]. Interestingly, a commonality to both interventions is the suppression of the inflammatory response. It is tempting to speculate that, regardless of the upstream event causing inflammation, blunting this response might ameliorate pathological symptoms of mitochondrial disease models. Our results indicate that a decrease in NADPH and GSH levels, associated with increased ASK1-p38-JNK signaling, contributes to the oxidative stress and inflammation in CI-defective cells. Targeting components of this metabolic/signaling cascade could be therapeutically exploited in mitochondrial diseases.

## Methods

**Cell culture and treatments.** All cell lines were maintained in Dulbecco's modified Eagle's medium (DMEM) high glucose (HyClone), 10% fetal bovine serum

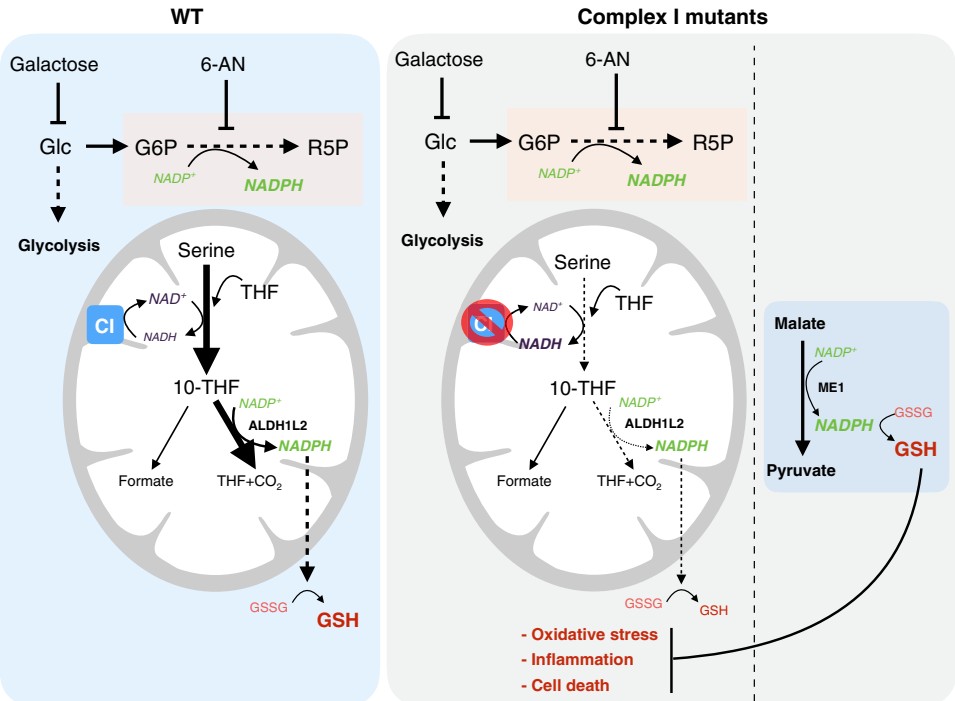

**Fig. 6 Model depicting how inhibition of PPP-generated NADPH is compensated in WT cells by enhancing mitochondrial one-carbon metabolism and induction of ALDH1L2.** Cells with defects in complex I displayed reduction of serine catabolism and concomitant NADPH production, which reduced GSH levels and increased oxidative stress, inflammation, and cell death. ME1 overexpression is able to rescue redox imbalance by acting as a potent source of NADPH and represents a potential therapeutic target to treat disorders associated with ETC dysfunction.

(FBS), and 1% penicillin/streptomycin (P/S) at 37 °C and 5% CO$_2$. MELAS and Rieske KO cells were cultured with supplementation of 50 μg/mL uridine. Homoplasmic human ND1 mutant and control cybrids were obtained from Rutger Vogel and Jan Smeitink Radbound University Medical Centre, Netherlands. Homoplasmic LHON ND6 cybrid cells and mouse fibroblast with deletions in ND4 (delA10227) were a gift from Carlos Moraes, University of Miami and Jose Antonio Enriquez, CNIC, respectively. All cell lines were tested routinely, and before all metabolomic analyses, for mycoplasma contamination. GSH reduced ethyl ester (GSH-MEE) (G1404), NAC (A9165), 6-aminonicotinamide (A68203), and Piericidin A (P4368) were purchased from SIGMA.

**CRISPR activator library design and production**. The human CRISPR activation library (Calabrese P65-HSF) was designed and created by the Broad Institute as previously described[18].

**Lentiviral-mediated CRISPR/Cas9 targets**. Transfections were performed according to the manufacturer's instruction using the polyfect reagent (Qiagen, 301107). Guide sequences for CRISPR/Cas9 were cloned in the lenti-CRISPR v2 vector (Addgene, 52951), dCAS-VP64_Blast (pXPR_109) (Addgene, 61425), or pXPR_502 (Addgene, 96923). The following constructs were used: activator guides for sgME1#1, 5-CTGCGAGAAGCGCTGAGTCA-3; sgME1#2, 5-GCGAGAGG GGGTGGACGCGT-3; sgPRDX1#1, 5-GACTCGGCGGCTTTCCCTCGT-3; sgPRDX1#2, 5-GGGAAAGCGCCGAGTCATTC-3. KO guides for sgME1#1, 5-AT AGGACTTGGCCTTTACCC-3; sgME1#2, 5-ATAGAGCCAATTTACCCACA-3; sgSHMT1#1, 5-ACCACTCACAAGACCCTGCG-3; sgSHMT1#2, 5-AGGCCCAT GATGCGCCCATG-3; sgSHMT2#1, 5-GTAGTCAATGAGGCGAGCAT-3; sgSHMT2#2, 5-GTTGTTCAGACAGGACCCCA-3; sgALDH1L2#1, 5-TGAA-CACCCCTACTACTCGG-3; sgALDH1L2#2, 5-ATAATTGATAGTCCAAAGCA-3. ME1 overexpression plasmid (Addgene, 49163).

**Genome-wide CRISPR activator screen**. In all, $8.5 \times 10^7$ human ND1 mutant (3796A>G) cybrid cells, stably expressing dCas9-VP64, were seeded in a total of 28 150 mm × 25 mm dishes ($3.0 \times 10^6$ per dish). Cells were infected with the lenti-pooled library to achieve a 30–50% infection efficiency, corresponding to a multiplicity of infection of ~0.3–0.5. Media was changed 24 h later, and 0.5 μg/mL puromycin was added 48 h later and selected for 7 additional days. Cells were trypsinized and separated into 150 mm × 25 mm dishes ($4.0 \times 10^6$ per dish). Next day, cells were washed twice with phosphate-buffered saline (PBS), and media was changed to non-glucose DMEM with glutamine (4 mM) supplemented with 10 mM galactose (Sigma, D7050), 10% FBS, 1% P/S, and 1 mM pyruvate. Cells were cultured for 5 days in galactose media. After 5 days, galactose media was replaced

with high-glucose DMEM in order to expand the remaining cells. This process was repeated for a second time, and surviving cells were collected and genomic DNA was isolated using the QIAamp DNA Mini Kit (QIAGEN, 51304). Two biological replicate experiments were performed.

**Immunoblot**. Cells were harvested in RIPA buffer (10 mM Tris-HCl pH 8.0, 1 mM EDTA, 1% Triton X-100, 0.1% sodium deoxycholate, 0.1% sodium dodecyl sulfate, 140 mM NaCl, 1× protease inhibitor cocktail, 1 mM phenylmethanesulfonyl-fluoride), and proteins were quantified using the BCA assay (Pierce, 23228). The following antibodies were used for western blot analysis: Anti-ME1 antibody (ab97445), β-Actin (CTS8457), Anti-IDH1 (ab81653), Anti IDH2 (ab131263), Anti-FLAG (F3165), Anti-NDUFS2 (ab192022), Anti-UQCRC2 (ab14745), Anti-MTCOI (ab14705), Anti-SDHA (ab1715), Anti-SHMT1 (ProteinTech, 14149-1-AP), Anti-SHMT2 (ProteinTech, 11099-1-AP), Anti-ALDH1L2 (ProteinTech, 21391-1-AP), Anti-MTHFD2 (ProteinTech, 12270-1-AP), Phospho-ASK1 (Thr845) (CTS3765), ASK1 (CTS3762), Phospho-p38 MAPK (Thr180/Tyr182) (CTS4511), p38 MAPK (CTS9212), Phospho-SAPK/JNK (Thr183/Tyr185) (CTS9251), SAPK/JNK (CTS9252), Anti-NDUFS4 antibody (ab87399).

**Gene expression**. Total RNA was isolated with Trizol (Invitrogen, 15596-026). Two micrograms of RNA were used to generate complementary DNA (cDNA) with a High Capacity cDNA Reverse Transcription Kit (Applied Biosystems, 4368813) following the manufacturer's protocol. For gene expression analysis, cDNA samples were mixed with Sybr Green qPCR mastermix (Applied Biosystems, 4309155) and were analyzed by a CFX 384 Real-Time system (Bio-Rad). Primer sequences: Human: c-FOS F-ATCTGCAGCGAGCATCTGAG, R-GGAT GACGCCTCGTAGTCTG; GDF15 F-GACCCTCAGAGTTGCACTCC, R-GCCT GGTTAGCAGGTCCTC; IL-6 F-CCTGAACCTTCCAAAGATGGC, R-TTCAC CAGGCAAGTCTCCTCA; MCP1 F-GCTGTGATCTTCAAGACCATTG, R-TGGAATCCTGAACCCACTTCTG; PTGS2 F-CTGGCGCTCAGCCATACAG, R-CGCACTTATACTGGTCAAATCCC; TNFα F-CACAGTGAAGTGCTGGCAAC, R-AGGAAGGCCTAAGGTCCACT; ME1 F-GAGTGCTGACATCTGACATTGA, R-TTGGCTTCCGAAACACCAAAC; ME2 F-ATATACACCGACGGTTGGTCT, R-CATCAGTCACTACAACAGCCTT; ME3 F-TGAAGAAGCGCGGATACGA TG, R-GAAAGCAGGGCGGGATTAGG; PRDX1 F-CCACGGAGATCATTGCT TTCA, R-AGGTGTATTGACCCATGCTAGAT. Mouse: c-FOS F-GTGAAGACC GTGTCAGGAGG, R-GATCTGTCTCCGCTTGGAGT; IL-6 F-CTGCAAGAGAC TTCCATCCAG, R-AGTGGTATAGACAGGTCTGTTGG; MCP1 F-AGCTGTA GTTTTTGTCACCAAGC, R-TGCTTGAGGTGGTTGTGGAA; PTGS2 F-CATC CCCTTCCTGCGAAGTT, R-CATGGGAGTTGGGCAGTCAT; TNFα F-CATCT TCTCAAAATTCGAGTGACAA, R-TGGGAGTAGACAAGGTACAACCC.

**Oxygen consumption**. In intact cells: $1.0 \times 10^5$ of the indicated cell type were seeded in an XFe-24 Seahorse plate (Seahorse Biosciences, 102340) and allowed to adhere for 24 h at 37 °C and 5% $CO_2$. Medium was then removed, and cells were washed with pre-warmed unbuffered DMEM without bicarbonate (Sigma, D5030) supplemented with 15 mM glucose, 2 mM sodium pyruvate, and 1 mM glutamine. After the wash, 600 µL of the same buffer was added and cultured at 37 °C in a non-$CO_2$ incubator for 1 h. The Seahorse 24 optical fluorescent analyzer cartridge was prepared in the interim by adding 5 µM oligomycin, 0.5 µM FCCP, and 2 µM rotenone/5 µM antimycin A to each cartridge port. Oxygen consumption rates (OCRs) (pmol/min) were then measured for each treatment condition at 37 °C using the Seahorse Bioanalyzer instrument. After measurement, media was removed and 20 µL of RIPA buffer was added and protein concentration using BCA (Pierce 23228) was measured to normalize OCR values.

In isolated mitochondria: To minimize variability between wells, mitochondria were first diluted 10× in cold 1× MAS buffer (70 mM sucrose, 220 mM mannitol, 10 mM $KH_2PO_4$, 5 mM $MgCl_2$, 2 mM HEPES, 1.0 mM EGTA, and 0.2 % (w/v) fatty acid-free bovine serum albumin (BSA), pH 7.2 at 37 °C). Stock substrates 0.5 M malic acid, 0.5 M pyruvic acid or 0.5 M succinate, and 0.2 mM ADP were subsequently diluted to the concentration required for plating. Next, while the plate was on ice, 50 µL of mitochondrial suspension (containing 25 µg of mitochondrial protein) was delivered to each well (except for background correction wells). The XF24 cell culture microplate was then transferred to a centrifuge equipped with a swinging bucket microplate adaptor and spun at $2000 \times g$ for 20 min at 4 °C. The Seahorse 24 optical fluorescent analyzer cartridge was prepared in the interim by adding 4 mM ADP and 10 µM rotenone to each cartridge port. After centrifugation, 450 µL of prewarmed (37 °C) 1× MAS + substrates pyruvate/malate (10 mM/2 mM) or succinate (10 mM) was added to each well. In the case of succinate-driven respiration, 100 µM rotenone was also added to the MAS buffer. The mitochondria were viewed briefly under a microscope at 20× to ensure consistent adherence to the well. The plate was then transferred to the Seahorse XFe/XF24 Analyzer, and the experiment initiated.

**Metabolomics**. WT and Ndufs4 KO mice, 45–50 days of age, were sacrificed, and tissues were snap-frozen immediately. Cells or 15 mg of pulverized brain frozen tissue were incubated with 1 mL chilled 80% high-performance liquid chromatography-grade methanol (Fluka Analytical). Cell mixture was incubated for 15 min on dry ice prior to centrifugation at $18,000 \times g$ for 10 min at 4 °C. Supernatant was retained, and remaining cell pellet was resuspended in 800 µL chilled 80% methanol and centrifuged. Supernatant was combined with the previous retention and was lyophilized using a SpeedVac (Thermo Fisher). Lyophilized samples were resuspended in 20 µL ultrapure water and subjected to metabolomics profiling using the AB/SCIEX 5500 QTRAP triple quadrupole instrument. Data analysis was performed using the GiTools software. $^{13}$C-labeled ([U-$^{13}C_5$]) glutamine was acquired in SIGMA (605166). Negative ion values for NADPH were selected as they produce more robust and reliable results. In this analysis, there are some current methodological limitations particularly with samples related to tissue collection.

**Blue native (BN) gel electrophoresis**. Digitonin at 4 g per g mitochondrial protein was used to solubilize the electron transfer chain complexes and 50 µg were applied and run on pre-cast 3–12% gradient BN gels (NativePAGE$^{TM}$ NOVEX Life Technologies) according to the manufacturer's instructions. After electrophoresis, the complexes were electroblotted onto polyvinylidene difluoride membranes and sequentially probed with specific antibodies.

**Measurement of NADPH/NADP$^+$ ratio**. NADP$^+$ and NADPH levels were individually measured in cell lysates by using the NADP/NADPH-glo$^{TM}$ Assay Kit (Promega) according to the manufacturer's instructions, and NADPH/NADP$^+$ ratio was calculated.

**Formate synthesis and $CO_2$ release in isolated mitochondria**. Mitochondria were resuspended in experiment buffer EB (137 mM KCl, 2.5 mM $MgCl_2$, 10 mM HEPES, 1 mg/mL BSA, pH 7.4), and formate production assays were performed in 105 µL of EB with 150 µg mitochondrial protein, 3 mM Pi, 1 mM serine, and 1 mM ADP, at 37 °C for 30 min as previously described[13]. Formate production was stopped by centrifuging the mixture at $8000 \times g$ for 10 min (4 °C) and removal of supernatant. Replicate wells of 40 µL supernatant were analyzed using a formate assay kit (Sigma), where one well did not contain enzyme and was used as a background NAD(P)H control. Piericidin A was used at 1 µM, and the corresponding vehicle control was 0.1% (w/v) dimethyl sulfoxide. When drug treatment was used, mitochondria were incubated with drugs for 2 min at 37 °C prior to addition of serine, Pi, and ADP.

To measure serine-induce $CO_2$ release, mitochondria were incubated in the same conditions described above plus the addition of 1 µCi of $^{14}$C serine (Perkin Elmer NEC827050UC). After 30 min, the reaction was stopped by adding 200 µL of 1 M perchloric acid. Next, 2-phenylethylamine-saturated Whatman paper was placed under the tube cap in order to trap radiolabeled $CO_2$ during overnight incubation. Finally, the Whatman paper was placed in scintillation liquid, and

radioactive counts were measured in a scintillation counter (Perkin Elmer, 2900TR).

**Proliferation and cell death quantification**. For cell number and proliferation, $1.0 \times 10^5$ cells were seeded in 6-well plates grown in DMEM high glucose, 10% FBS, and 1% P/S at 37 °C and 5% $CO_2$ or DMEM with no glucose but supplemented with 4 mM glutamine (HyClone) and 10 mM galactose (Sigma G0750), 10% FBS, and 1% P/S. Cells were incubated in galactose-containing media for the indicated times and were trypsinized and counted for trypan-blue exclusion using a hemocytometer (NanoEnTek, DHC-N01). For cell death quantification, propidium iodide (PI) cell exclusion assays were carried out as a measure of cell death. Cells were incubated with PI and analyzed by flow cytometry. Sub-G1 peak was selected as indicative of apoptosis.

**Glutamine oxidation**. In all, $5.0 \times 10^4$ sgNeg or sgME1 ND1 cells were seeded in a 24-well plate (Corning). Next day, media containing 0.5 µCi of $^{14}$C Glutamine (Perkin Elmer, NEC451050UC) was added, cells were incubated at 37 °C for 3 h, and the reaction was stopped by adding 200 µL of 1 M perchloric acid. Next, 2-phenylethylamine-saturated Whatman paper was placed under the tube cap in order to trap radiolabeled $CO_2$ during overnight incubation. Finally, Whatman paper was placed in scintillation liquid, and radioactive counts were measured in a scintillation counter (Perkin Elmer, 2900TR).

**ROS measurement**. 2′,7′-Dichlorofluorescin diacetate (DCFDA) assay was performed according to the manufacturer's instructions. Briefly, cells were incubated with 5 µM CM-$H_2$DCFDA (Invitrogen) for 30 min. Excess DCFDA was removed by washing the cells twice with PBS, and labeled cells were then trypsinized, rinsed, and resuspended in PBS and fluorescence was measured by flow cytometry.

**Proteomic analysis**. Mass spectrometric analysis were performed as previously described[15]. The mass spectrometric proteomics data have been deposited to the ProteomeXchange Consortium via the PRIDE[42] partner repository with the dataset identifier PXD018839.

**Ethics approval for mouse work**. All animal studies and procedures were conducted according to a protocol approved by the Institutional Animal Care and Use Committee (IACUC) at Beth Israel Deaconess Medical Center, IACUC RN-150D (IACUC protocol number: 112-2014). All the procedures that involves handling and use of mice in the experiments proposed in this grant application will be in strict accordance with the policies and guidelines established by the Beth Israel Deaconess Medical Center Animal Research Facility, which is an AAALAC accredited (Association for Assessment and Accreditation of Lab Animal Care) and PHS Assurance with Office of Laboratory Animal Welfare and complies with all federal, state, and local laws.

**Statistical analysis**. All statistics are described in figure legends. In general, for two experimental comparisons, a two-tailed unpaired Student's $t$ test was used. Three replicates per treatment were chosen as an initial sample size unless otherwise stated in figure legends. All western blot analysis were repeated at least three times. Statistical significance is represented by asterisks corresponding to $*p < 0.05$, $**p < 0.01$, or $***p < 0.001$. GraphPad Prism 7 was used for statistical analysis.

**Reporting summary**. Further information on experimental design is available in the Nature Research Reporting Summary linked to this paper.

## Data availability

The authors declare that the data supporting the findings of this study are available within the paper and its Supplementary Information files and will be available from the corresponding author upon reasonable request. Proteomics data have been deposited to the ProteomeXchange Consortium via the PRIDE[42] partner repository with the dataset identifier PXD018839. Source data is available as a source data file. All CRISPR-based screening data generated during this study has been uploaded to figshare (https://doi.org/10.6084/m9.figshare.12310118.v1).

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

## Acknowledgements

We thank all the members of the Puigserver laboratory for discussions regarding this project. This work was supported by the National Institute of Health, Grants RO1 CA181217 NCI, RO1 GM121452 NIGMS, and NIH 5 R01 DK089883-08 and Department of Defense CDMRP W81XWH-17-1-0216 to P.P. E.B. was supported in part by an EMBO postdoctoral fellowship and MDA Development Grant. E.A.P. was supported by NIHF30 (1F30DE028206-01A1). C.F.B was supported by F32GM125243. S.P.G. was supported by an NIH grant GM67945.

## Author contributions

E.B. and P.P. designed the study, interpreted the data, and wrote the manuscript. E.B., E.A.P., and C.F.B. performed the experiments. J.G.D. assisted with the CRISPR activator screen. S.P.G. and M.J. helped with proteomic analysis.

## Competing interests

The authors declare no competing interests.
