## [Peer Review File · Nature Communications]

Reviewers' comments:

Reviewer #1 (Remarks to the Author):

None

Reviewer #2 (Remarks to the Author):

The present manuscript seems to contain adequate discoveries for Nature Communications but this does not alleviate the need for the manuscript to support the main claims in the title and abstract and to, especially in contentious areas, use the most robust measurement methods.

A more complete treatment of my concerns can be found in the prior review.

A few comments on the revision:

1. I still do not see any evidence for "Defective NADPH production" in CI deficient cells. There are no measurements of oxPPP flux (even though these can be made simply, if not easily, using radioactive CO₂). Despite the focus on the folate pathway as an NADPH source, which is very interesting, radioactive or other measurements of this pathways activity are also lacking.
2. There is controversy over the mechanism by which SHMT2 impacts respiration, with genetic evidence suggesting that methylene-THF and not formyl-THF as the key metabolite (Morscher et al., Nature).
3. There is no logical way for formate to rescue respiration in SHMT2 KO cells, without restoring formyl-THF, and thus ALDH1L2 substrate. So there is no logical way presented by the authors for formate to restore respiration without restoring mitochondrial folate NADPH production.
4. The authors partially clarified their metabolite measurement methods, which pointed to the inadequacy of the current approach (multiple choices known to lead to NADPH and GSH artifacts). They also failed to explain how the initial freezing was done (where seconds are critical) or how the folate measurements were made given the lability of meTHF (Burgos-Barragan et al., Nature).
5. Some exploration of the priority of ME1 over PPP as rescue pathway is merited. Is this about substrate availability in the presence of galactose instead of glucose? the need to overexpress multiple enzymes to increase PPP activity (thus missed by the screen)? the utility of pyruvate (not sufficiency, but maybe pyruvate + more NADPH are needed)? would pyruvate + IDH1 over expression work?
6. It gets confusing in the paper that sgGENE seems to be used for both overexpression and under expression?
7. When certain results, like SHMT KD decrease growth, are galactose-specific, the authors should present the glucose results side-by-side in the main text. Results should also be given their simplest interpretation: When SHMT KO is known to impair respiration, and galactose is a respiratory substrate, we don't need a folate-NADPH explanation for these results.
8. Why don't the authors consider a title it reflects their main discovery, the rescue capacity of ME1?

Reviewer #4 (Remarks to the Author):

Although not many additional experiments were performed, we think the reply from the authors is adequate and addressed some of our major concerns. Given that we were already rather positive when we saw this paper for Nature, we feel this work would now be suited for publication.

Reviewer #5 (Remarks to the Author):

The revised manuscript is substantially improved.

Some minor comments:

- So the paper is accessible for the non-expert, sgRNA should be spelled out upon first use
- The use of "MELAS" in supplemental Figure 2 appears incorrect - MELAS is a clinical syndrome that can be caused by different mtDNA mutations, not just the A3242G mutation. The graph would be better labeled "A3243G".
- It is not immediately clear from the summary diagram in supplemental Fig. 3 how the intra-mitochondrial phenotype translates into inflammation and cell death. At the very least, it would be helpful to the reader to have an arrow with a question mark indicating that the mitochondrial state leads to cell death through an unknown pathway.

Point-by-point response to the reviewers' critiques

Reviewer 1

Overview-This is an interesting paper that uses an elegant positive CRISPR screen to pick out genes that protect against cell damage in cells with defective complex I. The main gene found was cytoplasmic malic enzyme which has the expected impact on cytosolic NADPH production and hence on cytosolic redox defense. The authors developed this into a nice story that related mitochondrial damage, one carbon metabolism, oxidative stress and the induction of inflammation. The take home message is important and fits well with the rest of the field. Much of the work is done well but there are technical points of concern that are discussed below. I have outlined some points that the authors may like to consider.

We thank the reviewer for the positive comments as well as the technical points indicated. In the revised manuscript, we have addressed these points with new experiments and explanations in the manuscript as well as in this response. In particular, we have addressed the compartmentalization of NADH and NADPH, and how this is dysregulated in mitochondrial disease CI mutant cells.

Major points

1 The links between the cytosolic and mitochondrial NADPH/NADP and NADH/NAD ratios are central to the paper but were not clarified. These ratios were assessed by methods which only look at whole cell levels. As the links between metabolism in the two compartments are key, more work on this is required. It would be good to assess changes in the redox state of these pools directly by fluorescence in cells and also by the use of fluorescent proteins that report on the NADH and NADPH pools (e.g. Nat Methods 2017 14 720).

We agree with the reviewer in this point that the link between the cytosolic and mitochondrial NADPH/NADP and NADH/NAD ratios explain the survival phenotypes of CI deficient cells. As suggested by the reviewer we have tried to obtain fluorescent sensor proteins, from the authors of the indicated publication, to assess compartmentalize changes in NADP/NADPH. Unfortunately, we have faced difficulties obtaining these reagents due to material transfer agreements that are still undergoing and they won't be available in the timeframe of the review process. However, to address and further clarify this link we have performed new experiments (new **Supplementary Fig.3d, e and f**) showing that the enzymes IDH2 (mitochondrial) and IDH1 (cytosolic) are required to maintain survival in wild-type cells indicating NADPH transfer from mitochondria to the cytosol using this shuttle. This transfer is deficient in CI or mitochondrial one-carbon mutant cells and expression of cytosolic ME1 is sufficient to rescue survival in these cells. This is also consistent with the fact that in new experiments we have expressed the NADH oxidase from *Lactobacillus brevis* (LbNOX) that localizes to the mitochondria and reduces NADH in this organelle. Importantly, mitochondrial LbNOX rescues survival of CI deficient cells (**Supplementary Fig.2c**). These experiments are now clarified in the model **Fig.6** and **Supplementary Fig.3f** (see also point 7 for further experimental details).

2 As related to point 1 above, the link between NADPH levels and antioxidant defense in the two compartments is key to interpreting these data. One concern is that the authors focus on the relationship between NADPH and the glutathione/glutathione disulfide pool through glutathione reductase. While glutathione plays a role, it is the action of NADPH via reductases 1 and 2 that sets the reduction potential of the mitochondrial and cytosolic thioredoxin pools that is more important, acting on peroxiredoxins and on thiol redox state to affect redox signalling. This is supported by the finding that Prx1 was one of the proteins that upon upregulation was protective. This area should have been assessed using redox active fluorescent proteins that report on local thiol redox state in the mitochondria and the cytosol.

We again agree with the reviewer. Our experiments clearly show that the reduction in GSH levels is underlying the cell death phenotype in mitochondrial CI mutant cells under conditions of inhibited PPP. This is consistent with the fact that GSH treatment rescues the cell death phenotype. The conclusion that NADPH deficiency in the cytosol, originated from decreased mitochondrial one carbon metabolism, triggers cell death is also supported by the fact that Prdx1 (cytosolic enzyme) rescues cell death and that ASK1 (cytosolic enzyme) is activated through phosphorylation that depends on oxidation of thiol groups¹.

3 The report of the NADPH/NADP and GSH/GSSG ratios as relative changes is OK, but not ideal compared to reporting the actual ratios. At least the ratios in the WT/control should be reported in the figure legends. Similarly,

measurements of ATP alone is not that useful. Far more meaningful to report ATP/ADP ratios. In addition, as these ratios are an average of several cell compartments - notably the mitochondrial and cytosolic – these points would be clarified and the limitations discussed. In my opinion it would be best to address these directly by the kind of experiments suggested in 1 and 2 above.?

As indicated by the reviewer, we have now included the relative ratios for NADPH/NADP, GSH/GSSG and ATP/ADP for WT samples, expressed as average \pm standard deviation (**Fig. 5e legends**). In addition, we have clarified that these are total cell values that include the different compartments. We have expressed ATP as the ratio ATP/ADP and consistent with our previous analysis no significant changes are observed (**Figure below**).

Metabolomic analysis in brain samples of WT and *Ndufs4* KO mice denoting changes in ATP/ADP ratio.

4 The data suggesting that there is activation of the reductive carboxylation of oxoglutarate by ICDH in these cells. But this will be done by NADPH-dependent ICDH, implying that there will be a mitochondrial NADPH pool. This should be discussed. The reductive carboxylation pathway generates citrate that is then exported to the cytosol by the citrate transporter and then broken down by ATP citrate lyase, so I would expect that this would be upregulated and that deletion of these would also be worth exploring.

We have specifically addressed this point with the experiments suggested by the reviewer (see previous point 1, and next 7) and discussed in the manuscript.

5 NAC is widely used as an antioxidant, but its mechanism of action is often obscure. If it is claimed to act by increasing cellular GSH levels then this has to be shown. Other mechanisms such as by increasing H₂S production (Ezerin et al., 2018, Cell Chemical Biology 25, 447–459) , or by its effects in the cell culture medium also need to be considered. Similarly, the use of the cell permeable ethyl GSH ester is fine, but it needs to be confirmed that it is actually increasing cell GSH levels.

We agree with the reviewer that the NAC mechanism of action is obscure. Based on the fact that it partially rescues the phenotype, in the manuscript we indicate potential differences beyond affecting GSH levels. We have confirmed that intracellular levels of GSH are increased in CI mutant cells when supplemented with 2mM of glutathione reduced ethyl ester (GSH-MEE). See **Figure below**.

GSH/GSSG ratio after supplementation with 2mM of Glutathione reduced ethyl ester measured by LC-MS.

6 Several other genes were turned up in the screen in Fig 1b, to nearly the same extent at malic enzyme. Some of which were clearly antiapoptotic (Bcl2 etc) but others were less clear. I assume that these are to be followed up in future work, but some discussion of how these may link to NADPH depletion would be useful. We are currently investigating other hits from the screen, but we think this goes beyond the scope of this publication. We think that discussing some of the hits might be too speculative, as the potential link at this point is unclear, and it will extend and diffuse the current focused discussion.

7 There was some confusion in my mind as to how mitochondrial one carbon metabolism and thus loss of NADPH in that compartment affected NADPH in the cytosol. The data seems to indicate that the production of NADPH in the mitochondrial matrix by ALDH1L2 was protective against loss of the PPP NADPH production in the cytosol, but how does the NADPH in the matrix how does this get to the cytosol? It obviously can't be NADPH itself but it may be other metabolites? This suggests that there are links that are not understood (or that I've missed something...).

We apologize for the initial confusion in the manuscript. As discussed in point 1, we were also puzzled on how mitochondrial NADPH is transferred to the cytosol in glucose restriction conditions when comparing wild-type and CI mutant cells. We completely agree with the reviewer that this a very relevant question in these studies. NADPH is a compartmentalized molecule unable to freely diffuse through lipid membranes. Within the TCA cycle, NADPH is a substrate for the mitochondrial NADP⁺-dependent IDH2 to reductively carboxylate αKG to isocitrate that can be converted to citrate and be exported to cytosol. Cytosolic isocitrate can then be oxidatively decarboxylated by cytosolic NADP⁺-dependent IDH1, producing cytosolic NADPH that is used to reduce glutathione disulfide. Interestingly, in our proteomic analysis (and by western blot) we observed that IDH1, and to a lower extent IDH2, were markedly upregulated upon galactose treatment (**Supplementary Fig.3c**). Genetic ablation of IDH1 sensitized wild type cells to galactose media and were rescued by GSH supplementation (**Supplementary Fig.3d**). Furthermore, similar to CI or mitochondrial one carbon mutant cells, we observed glucose restriction-induced cell death when IDH1-R132H and IDH2-R172K mutant enzymes were overexpressed in wild type cells. These mutations change the function of the enzymes, causing them to produce 2-hydroxyglutarate at expense of consuming NADPH during the process. These biochemical reactions create a futile cycle where WT-IDH1/2 produced NADPH is downstream exhausted by mutant-IDH1/2 (**Supplementary Fig.3e and figure below**). This indicates that mitochondrial NADPH shuttle system involving isocitrate dehydrogenases IDH2 and IDH1 could be responsible for transferring NADPH from the mitochondria to the cytosol (**Supplementary Fig.3f**). This mechanism provides additional reducing power that alleviates cytosolic oxidative stress in conditions of diminished PPP activity.

Model proposing how IDH2 and IDH1 can work in synchrony to transfer ALDH1L2 generated NADPH from the mitochondria to the cytosol. When ectopically expressed IDH1 and IDH2 mutants create a futile cycle consuming WT-IDH1/2 produced NADPH (see text for further details).

8 A key and potentially serious oversight with the paper is that the mtDNA mutant load of the cybrids and fibroblasts was not discussed at all. Assuming that the cells were not homoplasmic for the mutations, then

imposing a galactose selection step is likely to shift the heteroplasm. The authors need to measure how mtDNA mutation load shifts during their interventions.

All the experiments have been performed using homoplasmic cybrid cells. This is now clarified in the Methods section.

Minor points

1 The redox terminology used in this paper was sometimes confusing. There is no such thing as “reduced glutathione” – there is glutathione (GSH) and glutathione disulfide (GSSG). Also the discussion in several places using reduction of refer to a decrease when talking about redox couples can be problematic. I find it better to reserve reduction for a redox reaction in contrast to oxidation and use terms like decrease/diminution/lowering etc to avoid misunderstandings.

We apologize for the confusing terminology. It has been corrected in the text.

2 There is a mitochondrial malic enzyme – were there any changes in this?

There are 2 mitochondrial malic enzymes ME2 and ME3. We have confirmed that they do not change under galactose conditions and they were not positive hits of the screen.

mRNA analysis in WT and ND1 cells under galactose conditions showing no significant differences in ME1, ME2 or ME3 expression.

3 The role of inflammation following the oxidative stress associated with disrupted NADPH production following complex I defects is interesting and fits with current expectations. The mechanism was not explored in details and the link to the release of mtDNA was speculative at this stage without more data.

We agree with these two reviewer’s points. Our results suggest that defects in NADPH production and GSH are sensed by ASK1 that upon phosphorylation triggers an inflammatory response through activation of p38 and JNK. ASK1 is activated by oxidative stress involving oxidation of specific thiol groups¹.

Release of mtDNA is linked to defects affecting mtDNA stability such as the case of TFAM KO cells² and is just added in the context of the discussion as another mechanism of inflammation.

Reviewer 2

We thank the reviewer for the detailed and insightful critiques. In the revised manuscript, we have addressed these critiques (both from the first and second submission) with new experiments and explanations that have strengthened these studies. In particular, we have performed new experiments to address the reviewer’s concerns **1**) mitochondrial NADPH production deficiency in CI mutant cells. We have directly measured NADPH production in mitochondria through the one-carbon metabolism using radioactive [3-¹⁴C] serine. Consistent with our results, NADPH production was increased in glucose restricted conditions in wild-type cells, but was strongly reduced in CI mutant cells; **2**) contribution of IDH1. Based on our new findings that IDH1 and IDH2 are increased during glucose restricted conditions, we have depleted these enzymes using CRISPR editing (or using point mutants) in wild-type cells and become sensitive to cell death upon glucose deprivation. These results indicate that IDH1-dependent NADPH production is also necessary to maintain survival in cells (see below for detailed explanation).

Critiques (Second Submission)

1. I still do not see any evidence for "Defective NADPH production" in CI deficient cells. There are no measurements of oxPPP flux (even though these can be made simply, if not easily, using radioactive CO₂). Despite the focus on the folate pathway as an NADPH source, which is very interesting, radioactive or other measurements of this pathway's activity are also lacking.

We apologize for this confusion in the manuscript. We would like to clarify that "Defective NADPH production in CI deficient cells" refers to mitochondrial-produced NADPH, and this has been included and emphasized in the manuscript. Based on the metabolomics data (**Supplementary Fig.2b**) galactose media negatively impact glycolytic and PPP metabolites at the same extent in wild-type and CI mutant cells. Further supporting this conclusion, and requested by the reviewer, we have performed glucose tracing experiments, using fully labeled ¹³C-glucose, to follow the fate of glycolytic and PPP intermediates. In the **figure below**, both wild-type and CI mutant ND1 cells display similar levels of glycolytic and PPP intermediates with just small reduction at the level of 6-phosphogluconate. Increased levels of lactate might represent a compensatory mechanism to regenerate NAD⁺ in the cytosol by upregulating pyruvate to lactate conversion rate. These results indicate that the PPP fluxes (and derived NADPH production) are not significantly different between wild type and CI deficient cells.

WT and ND1 cells were incubated with fully labeled ¹³C-glucose for the indicated time points. Mass spectrometry analysis were performed focusing on the glucose-derived metabolites involved in the glycolytic and pentose phosphate pathway. No significant changes were observed.

As it relates to the folate pathway requested by the reviewer, we have performed new experiments to measure the contribution to NADPH production. Serine-derived 10-formyltetrahydrofolate is decarboxylated by the mitochondrial enzyme ALDH1L2 to tetrahydrofolate and CO₂ with equimolar production of NADPH. Based on this reaction, we have used an experimental approach to show that indeed mitochondrial NADPH production is defective in mitochondrial CI mutant cells. Similar to the formate assay (**Fig.4c**), isolated mitochondria (150ug) were incubated with radioactive ¹⁴C serine (1 uCi) and 1mM "cold" serine plus ADP (1mM) and Pi (3mM) for 30 mins at 37°C. The reaction was stopped by adding 200 uL of 1 M perchloric acid. Next, 2-phenylethylamine saturated Whatman paper was placed under the tube cap in order to trap radiolabeled CO₂ during overnight incubation. Finally, the Whatman paper was placed in scintillation liquid and radioactive counts (C.P.M) were measured in a scintillation counter (Perkin Elmer, 2900TR). We used ALDH1L2 CRISPR cells as a negative control. We observed a significant reduction of serine-derive CO₂ in ND1 cells indicating decreased production of NADPH (**Fig.4i**). Based on the strong reduction of CO₂ release in ALDH1L2 CRISPR cells we rule out the possibility that CO₂ might alternatively be produced from the glycine cleavage system. Moreover, previous studies have shown that increasing glycine levels actually impaired methylene-THF production due to reverse

flux through SHMT2³. In addition, in these studies, supplementation of glycine in serine free media decreased the NADPH/NADP⁺ ratio.

Altogether, these results are consistent with our model that mitochondrial one-carbon metabolism flux rely on ETC function and CI disease mutations compromise this pathway and therefore the ability to produce mitochondrial NAPDH.

We have further shown that mitochondrial imbalance NAD⁺/NADH ratio due to defective CI activity, compromise mitochondrial one carbon metabolism. We used the NADH oxidase from *Lactobacillus brevis* (LbNOX) which has been shown to regenerate NAD⁺ in cells with disrupted ETC by selectively consuming NADH⁴. Consistent with imbalanced NAD⁺/NADH ratio constraining the flux of mitochondrial one-carbon metabolism, mitochondrial-targeted LbNOX was able to restore serine-derived formate in CI mutant cells. This data is now included in the new version of the manuscript (**Supplementary Fig.2c**).

2. There is controversy over the mechanism by which SHMT2 impacts respiration, with genetic evidence suggesting that methylene-THF and not formyl-THF as the key metabolite (Morscher et al., Nature).

We acknowledge this controversy cited by the reviewer but it is important to recognize the literature supporting that formate supplementation rescue mitochondrial translation and therefore respiration in SHMT2 KO cells⁵. We would like to indicate that the precise metabolite that impacts SHMT2-dependent respiration does not directly affect our conclusions that formate, at concentrations that rescue respiration, does not prevent cell death in SHMT2 KO cells (see also next point).

3. There is no logical way for formate to rescue respiration in SHMT2 KO cells, without restoring formyl-THF, and thus ALDH1L2 substrate. So there is no logical way presented by the authors for formate to restore respiration without restoring mitochondrial folate NADPH production.

Our interpretation is that most of the formate, at the concentrations that have been used (1mM), is preferentially channeled towards mitochondrial protein translation and not being used to generate sufficient NADPH to prevent cell death. Based on this, it is conceivable that increasing doses of formate (>1mM) could provide some rescue. In fact, we have found that formate concentrations >5mM promote mild rescue in SHMT2 KO cells although not at the same magnitude that GSH (2mM) supplementation (**Figure below**, left panel). Furthermore, low doses of GSH (100-500uM) were more effective at rescuing galactose-mediated cell death when these SHMT2 KO cells were supplemented with 5mM formate (**Figure below**, right panel). These results suggest that at low concentrations formate is preferentially used to support mitochondrial translation (respiration) with little contribution to NADPH production, however high formate concentrations are sufficient to partially rescue cell death in SHMT2 KO cells.

Cell number of WT and sgSHMT2Δ cells cultured in galactose for 72 h and supplemented with increasing doses of formate. GSH treatment was used as a positive control (left panel). Cell number of formate treated cells were further supplemented with indicated doses of GSH (right panel).

Relevantly, we would also like to point out that ALDH1L2 is not considered the only step at what NADPH is generated, MTHFD2/L can potentially generate some, that would be consistent with our results that SHMT2 KO cells have the strongest phenotype compared to ALDH1L2 KO cells.

4. The authors partially clarified their metabolite measurement methods, which pointed to the inadequacy of the current approach (multiple choices known to lead to NADPH and GSH artifacts). They also failed to explain how the initial freezing was done (where seconds are critical) or how the folate measurements were made given the lability of meTHF (Burgos-Barragan et al., Nature).

We apologize for the partial clarification regarding the metabolite measurement methods. As it relates to the initial freezing, cells grown in 6-cm tissue culture dishes were rapidly washed with cold PBS, and metabolites were immediately extracted by the addition of 80% HPLC-grade methanol, pre-chilled at -80C. Metabolites were extracted for targeted tandem mass spectrometry (LC-MS/MS) profiling via selected reaction monitoring (SRM) with polarity switching on a 6500 QTRAP mass spectrometer (AB/SCIEX) as previously described⁶⁻⁸. We performed these measurements at the Mass Spectrometry Core at Beth Israel Deaconess Medical Center led by Dr. John M. Asara. Data were analyzed by calculating the Q3 peak areas using MultiQuant 3.0 software (AB/SCIEX). All of this information is now included in the methods section.

5. Some exploration of the priority of ME1 over PPP as rescue pathway is merited. Is this about substrate availability in the presence of galactose instead of glucose? the need to overexpress multiple enzymes to increase PPP activity (thus missed by the screen)? the utility of pyruvate (not sufficiency, but maybe pyruvate + more NADPH are needed)? would pyruvate + IDH1 over expression work?

We agree with the reviewer that the substrate availability is what underlies ME1 rescue under galactose conditions in CI mutant cells. These experiments have been carried out under galactose conditions (glucose-free) where PPP activity is vastly diminished due to compromised glucose availability. This is reflected in our metabolomic analysis where both glycolytic and PPP intermediates were significantly decreased. Taken this data into consideration, activating PPP would likely not produce substantial more NADPH since the main substrate of the pathway glucose, is not present, and galactose is poorly converted to PPP. Conversely, ME1 reaction do not rely on glucose and glutamine-derived malate appears to be sufficient to rescue survival in CI mutant cells. In fact, our metabolomics data showed that, in galactose conditions, almost 80% of the labelled malate comes from glutamine (**Fig.2b**).

As it relates to IDH1, we have shown that IDH1 is induced in galactose conditions (**Supplementary Fig.3c**), and even though pyruvate (1mM) was present, this was not sufficient to rescue galactose-induced cell death in CI mutant cells. Additionally, we present new data confirming that deletion of IDH1 sensitize wild-type cells to galactose induced cell death (**Supplementary Fig.3d**). Altogether, our model is consistent with that cells require at least two of the three main cellular sites of NADPH production: PPP, ME1 and IDH1, because single inhibition of these three components, similar to CI mutants, compromise cell survival in glucose restricted conditions. Importantly, we have shown that the NADPH produced in the mitochondria through the one-carbon metabolism is linked to the IDH2/IDH1 shuttle, to transfer this reducing equivalent from the mitochondria to the cytosol and causes reduction of GSSG to GSH (**Supplementary Fig.3c, d, e and f**). The fact that IDH1 did not score in the screen (performed in 1 mM pyruvate) is consistent with the dependency on mitochondrial IDH2 (reductive carboxylation reaction) that exhibits low activity in CI mutant cells because of impaired one carbon metabolism.

6. It gets confusing in the paper that sgGENE seems to be used for both overexpression and under expression?

We agree that this terminology might create some confusion. In the new version of the manuscript sgGENE refers to over-expression and GENE Δ to under-expression.

7. When certain results, like SHMT KD decrease growth, are galactose-specific, the authors should present the glucose results side-by-side in the main text. Results should also be given their simplest interpretation: When SHMT KO is known to impair respiration, and galactose is a respiratory substrate, we don't need a folate-NADPH explanation for these results.

We have not observed significant growth defects in SHMT1/2 CRISPR cells when culture in complete 25mM glucose DMEM media. We observed however that SHMT1 CRISPR cells proliferate a little slower in galactose conditions without any sign of cell death. Importantly, cell death phenotype under galactose conditions was specific to SHMT2 CRISPR cells. Based on our experiments (**Fig.4f and g**), the reason we favored an NADPH

explanation is because formate at lower concentrations was able to rescue respiration in SHMT2 KD but not cell survival (see above) suggesting that respiration is not sufficient. This interpretation is now included in the manuscript. We would also like to indicate that galactose is poorly metabolized and not being used as a substrate for respiration in these conditions, unless GALT enzymes are overexpressed, but they will be mainly used for glycolysis. In fact, removal of galactose, in glucose free-conditions, does not affect survival in cells. In these conditions, glutamine is the main substrate that is the main respiratory substrate.

8. Why don't the authors consider a title it reflects their main discovery, the rescue capacity of ME1?

We consider that the main discovery of this manuscript is that mitochondrial-produced NADPH becomes essential under nutrient stress conditions that constrain PPP activity, and this pathway is defective in mitochondrial disease CI deficient cells. In this context, ME1 activation bypasses this NADPH deficiency and strongly promotes survival in these mutant cells. Although we agree with the reviewer that the initial discovery is the rescue capacity of ME1, this main finding has allowed us to identify the blockage of mitochondrial one-carbon metabolism and NADPH production in CI deficient mutant cells.

Critiques (First Submission)

The present manuscript focuses on survival and growth of a particular “cybrid” complex I deficient cell type in galactose. Through a genome-scale overexpression screen, they identify malic enzyme 1 (cytosolic, NADPH producing) is a gene that enables cell growth under the galactose stress condition. Other hits of similar impact include BCL2 more members, suggesting that the growth inhibition/cell death arises from activation of apoptosis. Based on rescue by GSH and NAC but not MitoQ, it seems that cytosolic oxidative stress is a major trigger of these. Through Fig 2f, these are nice results that make a coherent and interesting story.

Two major suggestions:

1. It seems normative to demonstrate the ME1 effect across a diversity of cell types and stressors.

We agree with this reviewer’s point. In this manuscript, we have overexpressed ME1 in two different models of mild CI dysfunction as well as in three different wild-type cells where CI was inhibited pharmacologically (**Fig.1e and Supplementary Fig.1a,b,c and d**). In addition, we have also tested 3 different cell lines with strong inhibition of the electron transport chain and show that GSH rescue cell death (**Supplementary Fig.2a**). In addition, we have used a different stressor, antimycin A that selective inhibits CIII observing similar effect as with CI inhibitors (see graph below, and **Supplementary Fig.1d**). We think that these results support the conclusion that ME1 overexpression is an important mechanism that promotes survival in electron transfer chain compromised cells.

ME1 overexpression rescue cell death in U2OS cells after pharmacological inhibition of Complex I or Complex III using different doses of Piericidin or Antimycin respectively.

2. Some investigation into the “special” nature of ME1 versus IDH1 or oxPPP seems warranted. A rather straightforward hypothesis, which the authors do not explored, is that ME1 makes pyruvate which would itself likely rescue the cells (indeed is used clinically to ameliorate CI disorders).

We have tested pyruvate supplementation and only produced a very minor rescue in cell death, under galactose conditions, when added at 10mM or higher concentrations. In addition, the tracer experiments show that a large fraction of the ME1-produced pyruvate derives from glutamine (4 mM in the culture medium) suggesting that the main mechanism of ME1 is through NADPH production, which is consistent with the GSH rescue. This point is now addressed in large detail in the point 5 second critiques (see above).

The remainder of the paper lacks the coherence, accuracy or appeal to me of the first part. Fig 2h shows that ME1 can partially rescue a non-specific oxPPP inhibitor or H₂O₂. These are hardly novel findings, as oxPPP and ME1 are well known to be redundant pathways to NADPH which fights ROS.

Although we understand this point, to our knowledge, there have not been studies showing that mitochondrial CI mutant cells are dependent on PPP and ME1 because they cannot upregulate mitochondrial 1C metabolism. Importantly, PPP inhibition is well tolerated by WT cells but not by CI mutants. In addition, based on the consistent inflammatory phenotypes (and derived pathologies) observed in mitochondrial diseases, this manuscript mechanistically connects defects in mitochondrial 1C metabolism (see point 1, second critiques) linked to CI deficiency and inflammation.

Fig. 3 shows that CI deficiency somewhat impairs mitochondrial 1C flux, which is not unexpected given that such flux makes NADH, but does not get to the point of showing that this is functionally important. More critically, the connection to ME1 is lost, except for 1 piece of evidence that ME1 rescues SHMT2 loss, which may be cell background dependent since a comparable cell culture impaired growth phenotype for SHMT2 KO has not been seen in other contexts. Appropriate methods for the folate measurements are not provided-- this is critical as methylene-THF is quite unstable and normally polyglutamated in cells.

We would like to point out to the reviewer that there is a critical regulatory point to take into account. Our studies are designed within a nutrient stress context (galactose conditions) that requires mitochondrial respiration and protection against oxidative stress. In these conditions we show that maintaining NADPH balance, without changes in ATP or respiration, is required and sufficient to rescue cell death. We would also like to clarify that we do not claim that mitochondrial 1C metabolism is functionally connected to ME1 (**see model, Fig. 6**); it compensates for the deficiency of this pathway in CI mutant cells.

In this context, we also think is very relevant and important the fact that ME1 rescues SHMT2 loss. We don't think this rescue is cell background dependent, but it is mainly observed in galactose conditions. The key finding is that SHMT2 loss dependent cell death is not linked to a mitochondrial translation defect (and ATP deficiency) –please see formate experiments (**Fig.4f and g**), but to an NADPH defect and this is rescued by ME1 or GSH. In addition, and related to this point, in the new version of the revised manuscript we are including new data addressing how CI deficiency impairs mitochondrial 1C flux (**Supplementary Fig.2c**). We have used an innovative methodology employing the NADH oxidase from *Lactobacillus brevis* (LbNOX) that localizes to the mitochondria and decreases NADH levels. Importantly, since CI mutant cells accumulate NADH in the mitochondria, we now show that expression of this enzyme in the mitochondrial compartment is sufficient to restore one carbon metabolism flux in CI mutant cells.

A detailed explanation of how folate measurements were carried out can be found in point 4 (second critiques) that is included in the method section of the new version of the manuscript.

Fig.4 shows some inflammation data and metabolite levels in NDUSF4 KO mice. It seems that there are several unknowns in the metabolite measurement-- it's unclear how animal prepared and whether whole brain or part of brain was sampled, whether there was freeze clamp used, why solvents were chosen, whether metabolites are stable to the drying step (certainly GSSG can be made at this step). These are critical issues. In the spirit of openness, I should also admit that they weigh particularly heavily on me because we have also done metabolite analysis on NDUSF4 brains and see none of these changes.

We apologize for haven't explained in more detail the methodologies employed in these experiments and have been added in the new version of the manuscript and addressed in the point 4 (second critiques).

Even if these are all correct, it is still hard to see how the data support key claims of “Defective NADPH production” (title) (where is the evidence for defect in oxPPP or IDH1 or native ME1, the main cytosolic

pathways?) or that survival upon nutrient stress or oxPPP inhibition is dependent on NADPH production via folate pathway (abstract) (what experiment directly goes after this?).

In the new manuscript we have added new experiments showing that during nutrient stress conditions mitochondrial-produced NADPH becomes essential, and this is defective in mitochondrial disease CI mutant cells (See experiments detailed in point 1 and 5, second submission. In these conditions, our results show that ME1 activation bypasses this NADPH deficiency and strongly promotes survival in these mutant cells (see also point 8, second critiques).

Reviewer 4

Although not many additional experiments were performed, we think the reply from the authors is adequate and addressed some of our major concerns. Given that we were already rather positive when we saw this paper for Nature, we feel this work would now be suited for publication.

We thank the reviewer for the positive comments and indication that the manuscript is now suited for publication.

Reviewer 5

The revised manuscript is substantially improved.

We thank the reviewer for indicating the substantial improving of the first revision and the minor points requested have now been addressed.

Some minor comments:

- So the paper is accessible for the non-expert, sgRNA should be spelled out upon first use.

This is now included in the new version of the manuscript.

- The use of "MELAS" in supplemental Figure 2 appears incorrect - MELAS is a clinical syndrome that can be caused by different mtDNA mutations, not just the A3242G mutation. The graph would be better labeled "A3243G".

This has been modified as suggested by the reviewer and included in the new version of the manuscript.

- It is not immediately clear from the summary diagram in supplemental Fig. 3 how the intra-mitochondrial phenotype translates into inflammation and cell death. At the very least, it would be helpful to the reader to have an arrow with a question mark indicating that the mitochondrial state leads to cell death through an unknown pathway.

A new summary diagram has been designed incorporating the reviewer's suggestions.

References:

- 1 Nadeau, P. J., Charette, S. J. & Landry, J. REDOX reaction at ASK1-Cys250 is essential for activation of JNK and induction of apoptosis. *Mol Biol Cell* **20**, 3628-3637, doi:10.1091/mbc.E09-03-0211 (2009).
- 2 West, A. P. *et al.* Mitochondrial DNA stress primes the antiviral innate immune response. *Nature* **520**, 553-557, doi:10.1038/nature14156 (2015).
- 3 Fan, J. *et al.* Quantitative flux analysis reveals folate-dependent NADPH production. *Nature* **510**, 298-302, doi:10.1038/nature13236 (2014).
- 4 Titov, D. V. *et al.* Complementation of mitochondrial electron transport chain by manipulation of the NAD⁺/NADH ratio. *Science* **352**, 231-235, doi:10.1126/science.aad4017 (2016).
- 5 Minton, D. R. *et al.* Serine Catabolism by SHMT2 Is Required for Proper Mitochondrial Translation Initiation and Maintenance of Formylmethionyl-tRNAs. *Mol Cell* **69**, 610-621 e615, doi:10.1016/j.molcel.2018.01.024 (2018).
- 6 Yuan, M., Breitkopf, S. B., Yang, X. & Asara, J. M. A positive/negative ion-switching, targeted mass spectrometry-based metabolomics platform for bodily fluids, cells, and fresh and fixed tissue. *Nat Protoc* **7**, 872-881, doi:10.1038/nprot.2012.024 (2012).

- 7 Ben-Sahra, I., Howell, J. J., Asara, J. M. & Manning, B. D. Stimulation of de novo pyrimidine synthesis by growth signaling through mTOR and S6K1. *Science* **339**, 1323-1328, doi:10.1126/science.1228792 (2013).
- 8 Yuan, M. *et al.* Ex vivo and in vivo stable isotope labelling of central carbon metabolism and related pathways with analysis by LC-MS/MS. *Nat Protoc* **14**, 313-330, doi:10.1038/s41596-018-0102-x (2019).

REVIEWERS' COMMENTS:

Reviewer #1 (Remarks to the Author):

None

Reviewer #2 (Remarks to the Author):

In this high-quality revision, the authors effectively address my most important concerns. It is exciting to see progress from different directions in the field of NADPH biology, and the copious high-quality data in this paper will play an important role in the field going forward. I have the following OPTIONAL issues for the authors to consider prior to publication (without need for further review from me):

1. The fact that the key findings emerge at the confluence of ETC defects and glucose restriction could be made more prominent. It's hard to know how physiological this confluence is. Maybe it occurs rarely, but maybe it is actually central to mitochondrial diseases. Either way, it seems that most of the current findings are specific to glucose restriction, and making this abundantly clear to readers would be beneficial.
2. I appreciate the full details on the metabolomics methods. These indicate that the samples were quenched with 80% methanol (which is now recognized as insufficient to robustly capture NADPH) and subsequently dried (which can cause profound alterations in redox metabolites especially GSSG). The Authors may wish to remove the in vivo metabolomics (or the redox metabolite data portion). While I recognize that these data support their claims, I am suspicious of their robustness. Alternatively, the Authors may wish to indicate that these data should be interpreted with caution given the above methodological limitations.

Point-by-point response to the reviewers' critiques

(please see our response in red color)

Reviewer #1 (Remarks to the Author): None

Reviewer #2 (Remarks to the Author): In this high-quality revision, the authors effectively address my most important concerns. It is exciting to see progress from different directions in the field of NADPH biology, and the copious high-quality data in this paper will play an important role in the field going forward. I have the following OPTIONAL issues for the authors to consider prior to publication (without need for further review from me): **As this was a very critical reviewer, it is important to emphasize these overall remarks including high quality revision and how this manuscript will play an important role in the field going forward. We would like to add that, in our view, our studies go beyond NADPH biology and are important to understand the pathologies associated with mitochondrial diseases. Although the reviewer considers these remarks OPTIONAL we have edited the manuscript with the changes indicated below.**

1. The fact that the key findings emerge at the confluence of ETC defects and glucose restriction could be made more prominent. It's hard to know how physiological this confluence is. Maybe it occurs rarely, but maybe it is actually central to mitochondrial diseases. Either way, it seems that most of the current findings are specific to glucose restriction, and making this abundantly clear to readers would be beneficial. **We have emphasized now in the manuscript that the studies in cultured cells are performed under glucose restriction or inhibition of the pentose phosphate, both conditions limit cytosolic NADPH production. As it relates to the physiological context, we show data from the Ndufs4 KO mice that NADPH and glutathione levels are reduced in brain (Fig. 5e and Supplementary Fig. 4e) . These results suggest that the findings of this manuscript are central to mitochondrial diseases; however, they might go beyond and have implications in other pathologies associated with mitochondrial dysfunction such as ageing, diabetes, cancer or neurodegenerative disorders.**

2. I appreciate the full details on the metabolomics methods. These indicate that the samples were quenched with 80% methanol (which is now recognized as insufficient to robustly capture NADPH) and subsequently dried (which can cause profound alterations in redox metabolites especially GSSG). The Authors may wish to remove the in vivo metabolomics (or the redox metabolite data portion). While I recognize that these data support their claims, I am suspicious of their robustness. Alternatively, the Authors may wish to indicate that these data should be interpreted with caution given the above methodological limitations. **We have applied the same procedure for metabolite isolation and LC-MS-based measurement of metabolites including NADP, NADPH, GSH and GSSG through the whole length of these studies. We have observed strong robustness and consistency in our in vitro studies. Different challenges (glucose restriction or PPP inhibition) as well as multiple genetic perturbations (ME1, SHMT2) have consistently produced the same results; that NADPH and GSH are reduced in conditions of glucose limitation specificity in mitochondrial impaired cells. Hence, as the reviewer indicates our in vivo that data is entirely consistent with the in vitro data. Nevertheless, in the method section we have indicated the methodological limitations of these assays.**